# A UNIFIED FRAMEWORK OF THEORETICALLY ROBUST CONTRASTIVE LOSS AGAINST LABEL NOISE

## ABSTRACT

Learning from noisy labels is a critical challenge in machine learning, with vast implications for numerous real-world scenarios. While supervised contrastive learning has recently emerged as a powerful tool for navigating label noise, many existing solutions remain heuristic, often devoid of a systematic theoretical foundation for crafting robust supervised contrastive losses. To addresss the gap, in this paper, we propose a unified theoretical framework for robust losses under the pairwise contrastive paradigm. In particular, we for the first time derive a general robust condition for arbitrary contrastive losses, which serves as a criterion to verify the theoretical robustness of a supervised contrastive loss against label noise. This framework is not only holistic – encompassing prior techniques such as nearest-neighbor (NN) sample selection and robust contrastive loss – but also instrumental, guiding us to develop a robust version of the popular InfoNCE loss, termed Symmetric InfoNCE (SymNCE). Extensive experiments on benchmark datasets demonstrate the superiority of SymNCE against label noise.

## 1 INTRODUCTION

Supervised learning has demonstrated remarkable success across various machine learning domains, including computer vision (Krizhevsky et al., 2017; Redmon et al., 2016), information retrieval (Zhang et al., 2016; Onal et al., 2018), and language processing (Howard & Ruder, 2018; Severyn & Moschitti, 2015). However, it relies on clean and accurate labeled data. Unfortunately, real-world data is often noisy, with mislabeled or wrongly labeled data points, which can significantly degrade the performance of a supervised model. Consequently, learning with label noise becomes an important research problem in machine learning, and has been extensively studied in recent years.

Recently, supervised contrastive learning (Khosla et al., 2020) has been introduced to the problem of learning from label noise. Most methodological studies lean on the sample selection strategy, wielding contrastive learning as a mechanism to select confident samples based on the learned representations (Huang et al., 2023; Li et al., 2022; Ortego et al., 2021; Yao et al., 2021). For instance, Yan et al. (2022) leverage the negative correlations from the noisy data to avoid same-class negatives for contrastive learning. Navaneet et al. (2022) adopt nearest neighbor constraint to guarantee the high "purity" of the positive pairs, making the supervised contrastive learning approach robust to label noise. While these methods exhibit promising results, they often lack theoretical guarantees. On the other hand, some theoretical studies focus on the pretraining approaches, and prove the robustness of downstream classifiers with features learned by self-supervised contrastive learning (Cheng et al., 2021; Xue et al., 2022) without using noisy labels in pre-training stage. Chuang et al. (2022) propose the Robust InfoNCE (RINCE) loss against noisy views. Nonetheless, its theoretical analysis only applies to the specific proposed loss, and there is still a lack of theoretical research on the general robust condition for contrastive losses.

Therefore, we aim to establish a comprehensive theoretical framework for robust supervised contrastive losses against label noise. Specifically, we first derive the robust condition for arbitrary pairwise contrastive losses, which serves as a criterion to verify if a contrastive loss is robust or not. This general robust condition provides a unified understanding of existing robust contrastive methods and can inspire new robust loss functions. Based on our theory, we show that the widely used InfoNCE loss fails to meet the general robust condition, and we propose its robust counterpart

called **Sym**metric Info**NCE** (SymNCE) by adding a properly designed **Rev**erse Info**NCE** (RevNCE) to the InfoNCE loss function.

The contributions of this paper are summarized as follows.

- We for the first time establish a unified theoretical framework for robust supervised contrastive losses against label noise. We highlight that our theory serves as an inclusive framework that applies to existing robust contrastive learning techniques such as nearest neighbor (NN) sample selection and the RINCE loss (Chuang et al., 2022).

- Inspired by our theoretical framework, we propose a robust counterpart of the widely used InfoNCE loss function called SymNCE by adding InfoNCE with RevNCE, which is a deliberately designed loss based on our derived robust condition. RevNCE helps InfoNCE to be robust and meanwhile functions similarly as InfoNCE, i.e. aligning the positive samples and pushing away the negatives.

- We empirically verify that our SymNCE loss is comparable and even outperforms existing state-of-the-art robust loss functions for learning with label noise.

## 2 RELATED WORKS

### 2.1 LEARNING WITH LABEL NOISE

The major approaches of learning with label noise include robust architecture (Goldberger & Ben-Reuven, 2017; Han et al., 2018a; Yao et al., 2018), robust regularization (Lukasik et al., 2020; Pereyra et al., 2017; Wei et al., 2021), sample selection (Han et al., 2018b; Song et al., 2019; Yu et al., 2019), loss adjustment (Hendrycks et al., 2018; Patrini et al., 2017; Zhang & Sugiyama, 2021), and robust loss function (Ghosh et al., 2017; Ma et al., 2020; Wang et al., 2019; Zhang & Sabuncu, 2018). While robust architecture adds noise adaptation layer or dedicated architectures to the deep neural networks, robust regularization explicitly or implicitly regularizes the model complexity and prevents overfitting to the noisy labels. Sample selection adopts specific techniques or network designs to identify clean samples. Though these methodologies have exhibited empirical success, they often lean on intricate designs and can sometimes appear heuristic in nature. Loss adjustment modifies the loss function during training, whereas the effectiveness relies heavily on the estimated noise transition matrix.

Diverging from the aforementioned strategies, robust loss functions typically come with the assurance of theoretical robustness and effectiveness. For instance, Ghosh et al. (2017) first theoretically proved the general robust condition for supervised classification losses, and showed that the Mean Absolute Error (MAE) loss is robust whereas Cross Entropy (CE) is not. However, as MAE performs poorly on complex datasets, Zhang & Sabuncu (2018) generalized the MAE and CE losses, and proposed the Generalized Cross Entropy (GCE) loss by applying the Box-Cox transformation to the probabilistic predictions. Wang et al. (2019) proposed the Symmetric Cross Entropy (SCE) loss by combining the CE loss with the provably robust Reverse Cross Entropy (RCE) loss. Ma et al. (2020) showed that any supervised loss can be robust to noisy labels by a simple normalization, and proposed the Active Passive Loss (APL) by combining two robust loss functions.

### 2.2 SUPERVISED CONTRASTIVE LEARNING

Contrastive learning algorithms (Chen et al., 2020; He et al., 2020) were first proposed for self-supervised representation learning, where unsupervised contrastive losses pulls together an anchor and its augmented views in the embedding space. Khosla et al. (2020) extended contrastive learning to supervised training and proposed the Supervised Contrastive (SupCon) loss which takes same-class augmented examples as the positive labels within the InfoNCE loss. SupCon achieves significantly better performance than the state-of-the-art CE loss, especially on large-scale datasets.

**Supervised contrastive learning with label noise.** Recently, supervised contrastive learning has been introduced to solve weakly supervised learning problems such as noisy label learning (Huang et al., 2023; Li et al., 2022; Ortego et al., 2021; Yan et al., 2022; Yang et al., 2022). For example, Ortego et al. (2021) proposed the interpolated contrastive learning which adopts the interpolated samples in the contrastive loss to avoid overfitting to noisy labels, whose learned embeddings are then used to select clean samples through the nearest neighbor (NN) technique. Li et al. (2022) not

only selected confident samples for classifier training with the NN technique, but also mined positive pairs from the confident same-class samples for the training of contrastive representation learning. Huang et al. (2023) proposed the twin contrastive learning model to learn robust representations, where the wrongly labeled samples are recognized as the out-of-distribution samples through a Gaussian mixture model (GMM). Yan et al. (2022) leveraged the negative correlations from the noisy data to avoid same-class negatives for contrastive learning, whereas the positive sample selection procedure remains unsupervised. Yang et al. (2022) proposed to use supervised contrastive learning for semi-supervised learning, where the learned representations are used to refine pseudo labels.

**Robust supervised contrastive learning.** Aside from the empirical success of supervision contrastive learning with label noise, there are also works focusing on robust supervised contrastive learning (Chuang et al., 2022; Navaneet et al., 2022). Navaneet et al. (2022) constrained the same-class positives in contrastive learning within the $k$-nearest neighbors of the anchor sample. This method is empirically effective but lacks theoretical guarantees. Inspired by the robust condition for supervised classification losses in Ghosh et al. (2017), Chuang et al. (2022) proposed the robust InfoNCE (RINCE) loss function against noisy views which is proved to be a lower bound of Wasserstein Dependency Measure even under label noise. However, this theoretical analysis lacks generality to arbitrary contrastive losses.

To summarize, while supervised contrastive learning has demonstrated immense empirical promise in navigating label noise, the design and theoretical investigation of robust contrastive losses is still under-exploited. To fill in the blank, here we propose a unified theoretical framework for robust contrastive losses. We derive a general robust condition for arbitrary contrastive losses, inspired by which we further propose a robust counterpart of the widely used InfoNCE loss called SymNCE.

## 3 PROPOSED ROBUST CONDITION FOR CONTRASTIVE LOSSES

In this section, we first introduce some preliminaries for the mathematical formulation of the risks for contrastive losses and label noise. Then we propose the formal formulation of contrastive risks under the distribution of label noise. After that, we can derive a general robust condition for arbitrary contrastive losses, and discuss the relationship between our theoretical result and related works.

### 3.1 PRELIMINARIES

Suppose that random variables $X \in \mathcal{X}$ and $c \in [C] := \{1, \ldots, C\}$. Let the input data $\{(x_i, y_i)\}_{i \in [n]}$ be i.i.d. sampled from the joint distribution $\mathrm{P}(X, c)$. For $i \in [C]$, we denote the marginal distribution $\pi_i = \mathrm{P}(c = i)$, and denote class conditional distribution $\rho_i = \mathrm{P}(x|c = i)$. Under the noisy label distribution, we denote $\tilde{c} \in [C]$ as the random variable of noisy label, and let the noisy input data $\{(x_i, \tilde{y}_i)\}_{i \in [n]}$ be i.i.d. sampled from the joint distribution $\mathrm{P}(X, \tilde{c})$. For $i \in [C]$, we denote $\tilde{\pi}_i = \mathrm{P}(\tilde{c} = i)$ and $\tilde{\rho}_i = \mathrm{P}(x|\tilde{c} = i)$ as the noisy marginal and class conditional distributions, respectively. For notational simplicity, we denote $\boldsymbol{\pi} = (\pi_i)_{i \in [C]}$ and $\tilde{\boldsymbol{\pi}} = (\tilde{\pi}_i)_{i \in [C]}$.

#### 3.1.1 MATHEMATICAL FORMULATIONS OF SUPERVISED CONTRASTIVE RISK

We first generalize the supervised contrastive learning loss proposed in Khosla et al. (2020) to arbitrary contrastive losses $\mathcal{L}(x, \{x_m^+\}_{m=1}^M, \{x_k^-\}_{k=1}^K; f)$, where $x$ is the anchor sample, $\{x_m^+\}_{m=1}^M$ are $M$ positive samples, $\{x_k^-\}_{k=1}^K$ are $K$ negative samples, and $f : \mathcal{X} \to \mathbb{R}^d$ denotes the representation function. For notational simplicity, we write $\mathcal{L}(\boldsymbol{x}; f)$ instead in the rest of this paper.

For the mathematical formulations, we follow the CURL framework (Arora et al., 2019) for the mathematical formulation of supervised contrastive learning under clean labels (without label noise). Note that CURL uses the concept of latent classes to describe the distribution of positive pairs. In supervised contrastive learning, we naturally assume the latent classes to be the annotated classes, since positive samples are selected as those with the same labels.

The generation process of positive and negative samples is described as follows: (i) draw positive and negative classes $c_+, \{c_k^-\}_{k \in [K]} \sim \boldsymbol{\pi}^{K+1}$; (ii) draw an anchor sample $x$ from class $c^+$ with probability $\rho_{c^+} = \mathrm{P}(x|c = c^+)$; (iii) draw $M$ positive samples $\{x_m^+\}_{m=1}^M$ from class $c^+$ with probability $\rho_{c^+}$; and (iv) for $k \in [K]$, draw negative sample $x_k^-$ from class $c_k^-$ with probability $\rho_{c_k^-}$.

Then the corresponding risk for loss $\mathcal{L}(\boldsymbol{x}; f)$ is formulated as

$$\mathcal{R}(\mathcal{L}; f) = \mathbb{E}_{c^+, \{c_k^-\}_{k=1}^K \sim \boldsymbol{\pi}^{K+1}} \mathbb{E}_{x, \{x_m^+\}_{m=1}^M \sim \rho_{c^+}, \ x_k^- \sim \rho_{c_k^-}, k \in [K]} \mathcal{L}(\boldsymbol{x}; f). \tag{1}$$

### 3.1.2 LABEL NOISE ASSUMPTIONS

We model the generation process of label noise through label corruption. We assume that the label corruption process is conditionally dependent of the true label and independent of data features. To be specific, denote $\tilde{c} \in [C]$ as the random variable of noisy label, and we use $q_j(i) := \mathrm{P}(\tilde{c} = i | c = j, x)$ to denote the probability of true label $j \in [C]$ corrupted to label $i \in [C]$. Then according to the law of total probability, the posterior probability of noisy labels can be expressed as $\mathrm{P}(\tilde{c} = y | x) = \sum_{j=1}^C q_j(i) \mathrm{P}(c = i | x)$.

Next, we take the classic symmetric label noise assumption as an example to formulate label noise. This assumption is common and widely adopted in the community of learning with label noise (Ghosh et al., 2017; Ma et al., 2020; Natarajan et al., 2013; Wang et al., 2019). The discussions about asymmetric label noise can be found in the Appendix A.1.

**Assumption 3.1** (Symmetric label noise). *For noise rate $\gamma \in (0, 1)$, we assume that there holds $q_i(i) = 1 - \gamma$ and $q_j(i) = \gamma/(C - 1)$ for all $j \neq i$.*

In Assumption 3.1, we assume that the label is corrupted to a noisy label with probability $\gamma$ and remains clean with probability $1 - \gamma$, where a label has uniform probability of corrupting to any of the other labels.

## 3.2 SUPERVISED CONTRASTIVE RISKS UNDER LABEL NOISE

For supervised contrastive learning under label noise, the positive pairs are selected according to the (corrupted) annotated labels. Therefore, we combine the framework of supervised contrastive risk in Section 3.1.1 and the label noise assumptions in Section 3.1.2 to formulate the supervised contrastive risk under label noise.

The generation process of positive and negative samples under label noise can be described as follows: (i) draw noisy positive and negative classes $\tilde{c}_+, \{\tilde{c}_k^-\}_{k \in [K]} \sim \tilde{\boldsymbol{\pi}}^{K+1}$; (ii) draw an anchor sample $x$ from class $\tilde{c}^+$ with probability $\tilde{\rho}_{\tilde{c}+} = \mathrm{P}(x | \tilde{c} = \tilde{c}^+)$; (iii) draw $M$ positive samples $\{x_m^+\}_{m=1}^M$ from class $\tilde{c}^+$ with probability $\tilde{\rho}_{\tilde{c}+}$; and (iv) for $k \in [K]$, draw negative sample $x_k^-$ from class $\tilde{c}_k^-$ with probability $\tilde{\rho}_{\tilde{c}_k^-}$.

Finally, we can formulate the noisy risk for loss $\mathcal{L}(\boldsymbol{x}; f)$ as

$$\widetilde{\mathcal{R}}(\mathcal{L}; f) = \mathbb{E}_{\tilde{c}^+, \{\tilde{c}_k^-\}_{k=1}^K \sim \tilde{\boldsymbol{\pi}}^{K+1}} \mathbb{E}_{x, \{x_m^+\}_{m=1}^M \sim \tilde{\rho}_{\tilde{c}+}, \ x_k^- \sim \tilde{\rho}_{\tilde{c}_k^-}, k \in [K]} \mathcal{L}(\boldsymbol{x}; f). \tag{2}$$

## 3.3 ROBUST CONDITION FOR CONTRASTIVE LOSSES

Compared with the contrastive risk under clean label distribution $\mathcal{R}(\mathcal{L}; f)$ in equation 1, the noisy contrastive risk $\widetilde{\mathcal{R}}(\mathcal{L}; f)$ in equation 2 suffers from additional error because of label corruption. Specifically, under the noisy label distribution, the positive samples have the same annotated labels, but their true labels can be different. Aligning such noisy positive labels harms the representation learning procedure. Therefore, in this part, we first separate the additional risk caused by label corruption from the clean contrastive risk by explicitly deriving the relationship between clean and noisy risks. Then we propose the robust condition for contrastive losses.

In Theorems 3.2, we show the relationship between the noisy contrastive risk $\widetilde{\mathcal{R}}(\mathcal{L}; f)$ and the clean contrastive risk $\mathcal{R}(\mathcal{L}; f)$.

**Theorem 3.2.** *Assume that the input data is class balanced, i.e. $\pi_i = 1/C$ for $i \in [C]$. Then under Assumption 3.1, for an arbitrary contrastive loss $\mathcal{L}(\boldsymbol{x}; f)$, there holds*

$$\widetilde{\mathcal{R}}(\mathcal{L}; f) = \Big(1 - (C/C - 1)\gamma\Big)^2 \mathcal{R}(\mathcal{L}; f) + (C/C - 1)\gamma \cdot \Big(2 - (C/C - 1)\gamma\Big) \Delta \mathcal{R}(\mathcal{L}; f), \tag{3}$$

*where*

$$\Delta \mathcal{R}(\mathcal{L}; f) := \mathbb{E}_{c^+, \{c_m^+\}_{m=1}^M, \{c_k^-\}_{k=1}^K \sim \boldsymbol{\pi}^{M+K+1}} \mathbb{E}_{x \sim \rho_{c^+}, \ x_m^+ \sim \rho_{c_m^+}, m \in [M], \ x_k^- \sim \rho_{c_k^-}, k \in [K]} \mathcal{L}(\boldsymbol{x}; f). \tag{4}$$

From Theorem 3.2, we show that the noisy risk is a linear combination of the clean risk $\mathcal{R}(\mathcal{L}; f)$ and the additional risk $\Delta\mathcal{R}(\mathcal{L}; f)$. Specifically, when $\gamma = 0$, i.e. the labels are clean, the RHS of equation 3 degenerates to $\mathcal{R}(\mathcal{L}; f)$. Note that in the additional risk term $\Delta\mathcal{R}(\mathcal{L}; f)$, the anchor sample $x$ and positive samples $\{x_m^+\}_{m=1}^M$ are independently and uniformly sampled from the input distribution. The additional risk $\Delta\mathcal{R}(\mathcal{L}; f)$ represents the negative influence of label corruption, because no feature information can be learned by minimizing $\Delta\mathcal{R}(\mathcal{L}; f)$ and aligning such independent positive samples.

In noisy label learning, the goal is to optimize the clean risk $\mathcal{R}(f)$, whereas we can only achieve the noisy risk $\widetilde{\mathcal{R}}(f)$ since the clean distribution remains unknown. Nonetheless, if the additional risk $\Delta\mathcal{R}(\mathcal{L}; f)$ is a constant, then minimizing the noisy and clean risks results in the same optimal representation function $f$.

**Corollary 3.3.** *Assume that the input data is class balanced, and there exists a constant $A \in \mathbb{R}$ such that $\Delta\mathcal{R}(\mathcal{L}; f) = A$. Then under Assumption 3.1, contrastive loss $\mathcal{L}$ is noise tolerant if $\gamma < \frac{C-1}{C}$.*

In Corollary 3.3, we give the general condition for an arbitrary contrastive loss function to be noise tolerant. Comparing with Ghosh et al. (2017), who proposes the general robust condition for supervised classification losses aligning the sample label and its model prediction, our theoretical framework applies to losses under the pairwise contrastive paradigm. The two frameworks require the same noise rate $\gamma \le \frac{C-1}{C}$ under the same label noise assumption but require different "symmetric" condition for the loss functions. Specifically, by Ghosh et al. (2017), a supervised loss function $\mathcal{L}(g(x), y)$ is noise tolerant if $\sum_{c=1}^C \mathcal{L}(g(x), c)$ is a constant, where $g(x)$ is the model prediction of $x$. When the data is class-balanced, this condition indicates that the expectation of $\mathcal{L}(g(x), c)$ w.r.t. all classes is a constant, i.e. $\mathcal{L}(g(x), c)$ is "symmetric" over all classes. On the other hand, by Corollary 3.3, for a supervised contrastive loss $\mathcal{L}(\boldsymbol{x}; f)$, we require the contrastive loss to be "symmetric" over all positive samples.

### 3.4 THEORETICAL CHARACTERIZATION OF EXISTING APPROACHES

We highlight that our theoretical analysis in Section 3.3 serves as an inclusive framework that applies to nearest-neighbor (NN) sample selection, a widely used robust contrastive learning technique, and the robust InfoNCE (RINCE) loss proposed in Chuang et al. (2022).

#### 3.4.1 NN SAMPLE SELECTION UNDER THE UNIFIED THEORETICAL FRAMEWORK

In the NN sample selection, the positive samples in the supervised contrastive learning are selected as the same-class samples near to the anchor point in the embedding space. This technique is often used in noisy label learning algorithms to select confident samples that are believed to have correct annotated labels (Navaneet et al., 2022; Ortego et al., 2021; Li et al., 2022). These works discuss that samples usually have the same ground truth label as the semantically similar examples in a neighborhood, so it is reasonable to use NN techniques to select confident samples. In this part, we give an alternative theoretical explanation that NN techniques can reduce the gap between the additional risk $\Delta\mathcal{R}$ and constant values, thus making the loss function robust to label noise.

Under our theoretical framework, we take the widely used InfoNCE loss, i.e.

$$\mathcal{L}_{\text{InfoNCE}}(\boldsymbol{x}; f) := \frac{1}{M} \sum_{m \in [M]} -\log \frac{e^{f(x)^\top f(x_m^+)}}{e^{f(x)^\top f(x_m^+)} + \sum_{k \in [K]} e^{f(x)^\top f(x_k^-)}}, \tag{5}$$

as an example to demonstrate how the NN technique enables contrastive learning to be robust against label noise. According to the definition of the additional risk, we have

$$\lim_{M, K \to \infty} \Delta\mathcal{R}(\mathcal{L}_{\text{InfoNCE}}; f) - \log K = -\mathbb{E}_{x, x^+ \overset{\text{i.i.d.}}{\sim} \mathrm{P}_X} f(x)^\top f(x^+) + \mathbb{E}_{x \sim \mathrm{P}_X} \log \mathbb{E}_{x^- \sim \mathrm{P}_X} e^{f(x)^\top f(x^-)}. \tag{6}$$

By Jensen's Inequality, there holds

$$\mathbb{E}_{x'} f(x)^\top f(x') \le \log \mathbb{E}_{x'} e^{f(x)^\top f(x')}, \tag{7}$$

and accordingly $\lim_{M, K \to \infty} \Delta\mathcal{R}(\mathcal{L}_{\text{InfoNCE}}; f) - \log K \ge 0$ for all $f$.

Next, we discuss that the NN technique reduces the value of the LHS of equation 7, making $\lim_{M,K\to\infty} \Delta\mathcal{R}(\mathcal{L}_{\text{InfoNCE}}; f) - \log K$ close to the constant 0, and thus enabling the loss function to be noise tolerant according to Corollary 3.3. Mathematically, we formulate the InfoNCE loss with the NN technique as

$$\mathcal{L}_{\text{InfoNCE-NN}}(\boldsymbol{x}; f, t) := \frac{1}{|\mathcal{I}_N(x;t)|} \sum_{m\in\mathcal{I}_N(x;t)} -\log \frac{e^{f(x)^\top f(x_m^+)}}{e^{f(x)^\top f(x_m^+)} + \sum_{k\in[K]} e^{f(x)^\top f(x_k^-)}}, \quad (8)$$

where for a threshold parameter $t \in [-1, 1]$, we let $\mathcal{N}(x;t) := \{x^+ : f(x)^\top f(x^+) \geq t\}$ be the neighbor set of sample $x$, and denote $\mathcal{I}_N(x;t)$ as the index set of $\mathcal{N}(x;t)$. The threshold parameter $t$ ensures that only the positive samples sharing high similarity with the anchor sample are used to calculate the InfoNCE loss. Note that when $t$ is selected as the quantiles of $f(x)^\top f(x')$, the positive sample set $\mathcal{N}(x;t)$ contains exactly the nearest neighbors of $x$ in the embedding space. Then for the corresponding additional risk w.r.t. $\mathcal{L}_{\text{InfoNCE-NN}}$, we have

$$\lim_{M,K\to\infty} \Delta\mathcal{R}(\mathcal{L}_{\text{InfoNCE-NN}}; f, t) - \log K = -\mathbb{E}_{x\sim P_X}\mathbb{E}_{x'\in\mathcal{N}(x;t)}f(x)^\top f(x') + \mathbb{E}_{x\sim P_X}\log\mathbb{E}_{x'\sim P_X} e^{f(x)^\top f(x')}. \quad (9)$$

Because $\mathbb{E}_{x'\in\mathcal{N}(x;t)}f(x)^\top f(x') \geq \mathbb{E}_{x'\sim P_X}f(x)^\top f(x')$, for a given $f$, we can select a proper threshold parameter $t$ to make $\lim_{M,K\to\infty} \Delta\mathcal{R}(\mathcal{L}_{\text{InfoNCE-NN}}; f, t) - \log K = 0$, and thus $\mathcal{L}_{\text{InfoNCE-NN}}$ is noise tolerant.

### 3.5 RINCE UNDER THE UNIFIED THEORETICAL FRAMEWORK

The mathematical form of the RINCE loss function (Chuang et al., 2022) is

$$\mathcal{L}_{\text{RINCE}}(\boldsymbol{x}; f) = -(1-\lambda)e^{f(x)^\top f(x^+)} + \lambda\sum_{k=1}^{K} e^{f(x)^\top f(x_k^-)}, \quad (10)$$

where $\lambda > 0$ is a density weighting term controlling the ratio between positive and negative pairs. Although inspired by the symmetric condition (Ghosh et al., 2017), RINCE does not fit this theoretical framework designed for supervised losses that align the model prediction and label. By contrast, our proposed inclusive theoretical framework for contrastive losses can guarantee the robustness of RINCE against label noise from the risk consistency perspective. By definition, we have the additional risk $\Delta\mathcal{R}(\mathcal{L}_{\text{RINCE}}; f)$ w.r.t. $\mathcal{L}_{\text{RINCE}}$ as

$$\mathbb{E}_{x,x^+,\{x_k^-\}_{k=1}^K \overset{\text{i.i.d.}}{\sim} P_X} -(1-\lambda)e^{f(x)^\top f(x^+)} + \lambda\sum_{k=1}^{K} e^{f(x)^\top f(x_k^-)} = ((K+1)\lambda - 1)\mathbb{E}_{x,x'\overset{\text{i.i.d.}}{\sim} P_X} e^{f(x)^\top f(x')}. \quad (11)$$

According to Corollary 3.3, when $\lambda = 1/(K+1)$, we have $\Delta\mathcal{R}(\mathcal{L}_{\text{RINCE}}; f) = 0$, and thus $\mathcal{L}_{\text{RINCE}}$ is noise tolerant. That is, as a byproduct, our theory provides a more specific theoretical choice of the parameter $\lambda$ in RINCE.

## 4 THEORETICALLY INSPIRED SYMMETRIC CONTRASTIVE LOSS

Inspired by Corollary 3.3, we propose a robust contrastive loss SymNCE by directly modifying the InfoNCE loss function. We first propose a reverse version of InfoNCE called RevNCE, and then SymNCE is designed by adding RevNCE to InfoNCE.

### 4.1 REVERSE INFONCE (REVNCE)

We first argue that the InfoNCE loss function is non-robust. Recall that in equation 6, we have $\lim_{M,K\to\infty} \Delta\mathcal{R}(\mathcal{L}_{\text{InfoNCE}}; f) - \log K \geq 0$ for all $f$. That is, the InfoNCE loss function violates the general robust condition proposed in Corollary 3.3. Therefore, we seek to add a reverse term to the InfoNCE loss such whose additional risk equals exactly to $-\lim_{M,K\to\infty} \Delta\mathcal{R}(\mathcal{L}_{\text{InfoNCE}}; f)$ through all representation functions $f$, so as to guarantee the robustness of the total loss function.

Although this theoretical robustness can be trivially achieved by adding $-\mathcal{L}_{\text{InfoNCE}}$ or other similar terms to the original InfoNCE, this would destroy the learning procedure. Therefore, in the meanwhile, this reverse term is also required to function similarly as InfoNCE, i.e. aligning the positive samples and pushing apart the negative ones.

To find a proper "reverse" loss that meets the above requirements, we utilize the symmetric property of the additional risk. Specifically, note that in the definition of $\Delta\mathcal{R}(\mathcal{L}_{\text{InfoNCE}}; f)$, the positive and negative samples are i.i.d. generated. Therefore, we can exchange them to get a reverse form of $\Delta\mathcal{R}(\mathcal{L}_{\text{InfoNCE}}; f)$ without changing its value, i.e.

$$\lim_{M,K\to\infty} -\Delta\mathcal{R}(\mathcal{L}_{\text{InfoNCE}}; f) + \log K = \mathbb{E}_{x,x^-\overset{\text{i.i.d.}}{\sim}\mathrm{P}_X} f(x)^\top f(x^-) - \mathbb{E}_{x\sim\mathrm{P}_X} \log \mathbb{E}_{x^+\sim\mathrm{P}_X} e^{f(x)^\top f(x^+)}$$

$$= \mathbb{E}_{x,x^-\overset{\text{i.i.d.}}{\sim}\mathrm{P}_X} - \log \frac{\mathbb{E}_{x^+\sim\mathrm{P}_X} e^{f(x)^\top f(x^+)}}{f(x)^\top f(x^-)}. \tag{12}$$

In Proposition 4.1, we propose the reverse InfoNCE loss (abbreviated as RevNCE), and show that its limit form is exactly the same as equation 12. The proof is shown in Appendix A.2.

**Proposition 4.1.** *Define the RevNCE loss function as*

$$\mathcal{L}_{\text{RevNCE}}(\boldsymbol{x}; f) = \frac{1}{K} \sum_{k\in[K]} -\log \frac{\frac{1}{M}\sum_{m\in[M]} e^{f(x)^\top f(x_m^+)}}{e^{f(x)^\top f(x_k^-)}}. \tag{13}$$

*Then we have* $\lim_{M,K\to\infty} \Delta\mathcal{R}(\mathcal{L}_{\text{RevNCE}}; f) = \lim_{M,K\to\infty} -\Delta\mathcal{R}(\mathcal{L}_{\text{InfoNCE}}; f) + \log K$.

Moreover, similar to InfoNCE, the RevNCE loss aligns the positive samples and meanwhile pushes away the negatives. It differs from the InfoNCE loss function only in the summation of the positives and negatives.

## 4.2 Symmetric InfoNCE (SymNCE)

By adding together the InfoNCE and RevNCE loss functions, we propose the symmetric InfoNCE (SymNCE) as

$$\mathcal{L}_{\text{SymNCE}}(\boldsymbol{x}; f) := \mathcal{L}_{\text{InfoNCE}}(\boldsymbol{x}; f) + \mathcal{L}_{\text{RevNCE}}(\boldsymbol{x}; f). \tag{14}$$

By Proposition 4.1, we have $\lim_{M,K\to\infty} \Delta\mathcal{R}(\mathcal{L}_{\text{SymNCE}}; f) = \lim_{M,K\to\infty} \Delta\mathcal{R}(\mathcal{L}_{\text{InfoNCE}}; f) + \lim_{M,K\to\infty} \Delta\mathcal{R}(\mathcal{L}_{\text{RevNCE}}; f) = \log K$. Then by Corollary 3.3, $\mathcal{L}_{\text{SymNCE}}(\boldsymbol{x}; f)$ is noise tolerant.

Intuitively, $\mathcal{L}_{\text{SymNCE}}$ is noise tolerant because $\mathcal{L}_{\text{RevNCE}}$ selects high-confidence positive samples. Because log-sum-exp approximates the max function, we have

$$\mathcal{L}_{\text{RevNCE}}(\boldsymbol{x}; f) \approx \frac{1}{K} \sum_{k\in[K]} - \max_{m\in[M]} \left\{ f(x)^\top [f(x_m^+) - f(x_k^-)] \right\} + \log M, \tag{15}$$

that is, minimizing $\mathcal{L}_{\text{RevNCE}}(\boldsymbol{x}; f)$ is to optimize over the positive sample sharing the highest similarity with the anchor. Samples with high semantically similarity usually have the same ground truth label, so $\mathcal{L}_{\text{SymNCE}}$ is robust against label noise because $\mathcal{L}_{\text{RevNCE}}$ adds additional weights to the highly confident same-class positives.

As is argued in previous works, robustness itself is insufficient for empirical performance and robust losses can suffer from underfitting (Ma et al., 2020). Therefore, it is common to introduce an additional weight parameter $\beta \in [0, 1]$ to balance between accuracy and robustness (Chuang et al., 2022; Ma et al., 2020; Wang et al., 2019). Empirically, we use

$$\widehat{\mathcal{L}}_{\text{SymNCE}}(x_i; f, \beta) := \widehat{\mathcal{L}}_{\text{InfoNCE}}(x_i; f) + \beta \cdot \widehat{\mathcal{L}}_{\text{RevNCE}}(x_i; f), \tag{16}$$

where the empirical forms are

$$\widehat{\mathcal{L}}_{\text{InfoNCE}}(x_i; f) := -\frac{1}{|P(i)|} \sum_{p\in P(i)} \log \frac{e^{f(x_i)^\top f(x_p)/\tau}}{\sum_{a\in A(i)} e^{f(x_i)^\top f(x_a)/\tau}}, \tag{17}$$

$$\widehat{\mathcal{L}}_{\text{RevNCE}}(x_i; f) := -\frac{1}{|A(i)|-1} \sum_{a\in A(i)\setminus\{i\}} \log \frac{\frac{1}{|P(i)|}\sum_{p\in P(i)} e^{f(x_i)^\top f(x_p)/\tau}}{e^{f(x_i)^\top f(x_a)/\tau}}, \tag{18}$$

$P(i) := \{p \in A(i) : \tilde{y}_p = \tilde{y}_i\}$ is the index set of all same-class positives distinct from $i$, $A(i)$ is the index set of all augmented samples, and $\tau > 0$ is the temperature parameter.

## 5 EXPERIMENTS

### 5.1 PERFORMANCE COMPARISONS ON BENCHMARK DATASETS

We first conduct numerical comparisons on CIFAR-10 and CIFAR-100 benchmark datasets (Krizhevsky et al., 2009). The noisy labels are generated following standard approaches in previous works (Ma et al., 2020; Wang et al., 2019). The symmetric label noise is generated by flipping a proportion of labels in each class uniformly at random to other classes. The proportion of flipped labels equals to the noise rate $\gamma$. For asymmetric noise, we flip the labels within a specific set of classes. For CIFAR-10, flipping TRUCK $\rightarrow$ AUTOMOBILE, BIRD $\rightarrow$ AIRPLANE, DEER $\rightarrow$ HORSE, and CAT $\leftrightarrow$ DOG. For CIFAR-100, the 100 classes are grouped into 20 super-classes with each having 5 sub-classes. We then flip each class within the same super-class into the next in a circular fashion. We vary the noise rate $\gamma \in \{0.2, 0.4, 0.6, 0.8\}$ for symmetric label noise and $\gamma \in \{0.2, 0.4\}$ for the asymmetric case.

It is worth mentioning that our paper focuses on designing new robust loss functions with sound theoretical guarantees, which is one of the most important parts of learning from label noise. We believe it is not so fair to directly compare a robust loss function with such algorithms incorporating multiple heuristic techniques and without theoretical guarantees. Therefore, we compare our proposed SymNCE with both robust supervised classification losses and robust supervised contrastive losses. For supervised classification losses, we compare with Cross Entropy (CE), Mean Absolute Error (MAE) (Ghosh et al., 2017), Generalized Cross Entropy (GCE) (Zhang & Sabuncu, 2018), Symmetric Cross Entropy (SCE) (Wang et al., 2019), and Active Passive Loss (APL). For supervised contrastive losses, we compare with SupCon (Khosla et al., 2020) and Robust InfoNCE (RINCE) (Chuang et al., 2022). The training details can be found in Appendix A.3.

The results are shown in Table 1. We observe that our SymNCE shows mostly better empirical performances under both symmetric and asymmetric label noise across various noise rates. Compared with classification losses, SymNCE has significant advantages under high noise rates and asymmetric label noise. Compared with both classification and contrastive losses, SymNCE has larger performance gains on the more complex CIFAR-100 dataset.

Table 1: Performance comparisons with state-of-the-art robust losses.

| Dataset | Noise rate | Symmetric | | | | | Asymmetric | |
|---|---|---|---|---|---|---|---|---|
| | | 0% | 20% | 40% | 60% | 80% | 20% | 40% |
| CIFAR-10 | CE | 92.88 | 85.22 | 78.9 | 71.98 | 41.38 | 86.88 | 82.12 |
| | MAE | 91.32 | 89.28 | 84.85 | 78.19 | 41.46 | 81.3 | 56.77 |
| | GCE | 91.83 | 89.22 | 84.66 | 76.66 | 42.21 | 88.01 | 81.05 |
| | SCE | 92.97 | 89.48 | 83.57 | 77.6 | 55.58 | 89.08 | 82.46 |
| | APL | 91.21 | 88.9 | 82.08 | 78.48 | 52.04 | 88.39 | 81.92 |
| | SupCon | 93.07 | 87.1 | 80.47 | 61.8 | 55.6 | 90.08 | 87.26 |
| | RINCE | 86.17 | 85.26 | 83.15 | 80.65 | **80.32** | 84.92 | 84.27 |
| | SymNCE (Ours) | **93.12** | **89.81** | **85.32** | **80.89** | 60.74 | **91.0** | **88.28** |
| CIFAR-100 | CE | 64.39 | 47.21 | 37.30 | 27.25 | 15.12 | 49.16 | 36.29 |
| | MAE | 13.53 | 8.84 | 8.44 | 6.63 | 2.73 | 11.63 | 7.69 |
| | GCE | 58.52 | 54.16 | 47.27 | 35.65 | 20.25 | 53.79 | 34.60 |
| | SCE | 66.83 | 60.32 | 52.79 | 39.24 | 20.33 | 59.29 | 40.49 |
| | APL | 34.22 | 28.38 | 25.27 | 16.95 | 10.68 | 28.98 | 21.70 |
| | SupCon | 68.05 | 61.23 | 53.02 | 38.74 | 25.37 | 64.98 | 55.33 |
| | RINCE | 44.41 | 44.29 | 42.27 | 41.46 | 38.99 | 42.49 | 32.68 |
| | SymNCE (Ours) | **70.30** | **64.56** | **55.68** | **43.35** | **39.14** | **67.15** | **56.5** |

### 5.2 PERFORMANCE COMPARISONS ON REAL-WORLD DATASETS

Beside benchmark datasets, we also evaluate our proposed SymNCE on the real-world dataset Clothing1M (Xiao et al., 2015), whose noise rate is estimated to be around 40%. Following Li et al. (2020), we select a subset which contains 1.4k samples for each class from the noisy training data and report the performance on the 10k test data. In Table 2, where the best performance is marked in

**bold**, we show that our SymNCE outperforms both robust classification losses and robust contrastive losses on the real-world dataset Clothing1M.

Table 2: Real-data comparisons with state-of-the-art robust losses.

| CE | GCE | SCE | APL | MAE | SupCon | RINCE | SymNCE |
|-------|-------|-------|-------|-------|--------|-------|--------|
| 53.48 | 56.01 | 57.48 | 36.15 | 36.83 | 62.50 | 43.30 | **65.20** |

### 5.3 PARAMETER ANALYSIS

Recall that in Section 4.2, we introduce a weight parameter $\beta$ to our $\mathcal{L}_{\text{SymNCE}}$, which is the only parameter of our method and balances between the accurate term $\mathcal{L}_{\text{InfoNCE}}$ and the robust term $\mathcal{L}_{\text{RevNCE}}$. We here conduct analysis of the weight parameter $\beta$. In Figures 1(a) and 1(b), we plot the test accuracy of SymNCE with different weight parameters $\beta \in \{0.2, 0.6, 1.0\}$ under symmetric label noise $\gamma \in \{0, 0.2, 0.4, 0.6, 0.8\}$ and asymmetric label noise $\gamma \in \{0, 0.2, 0.4\}$ on CIFAR-100, where $\gamma = 0$ corresponds to the clean dataset without label noise. We show that the optimal $\beta$ increases as noise rate $\gamma$ enhances. Specifically, for symmetric label noise, when noise rate is low ($\gamma = 0$), the optimal $\beta = 0.2$, and when noise rate is high ($\gamma \in \{0.2, 0.6, 0.8\}$), the optimal $\beta = 1.0$. For asymmetric label noise, the optimal $\beta$ is 0.2 when $\gamma = 0$, whereas raises to 1.0 when $\gamma = 0.4$. This is because robust loss functions are designed to avoid overfitting to label noise, and thus can suffer from underfitting (Ma et al., 2020) when noise rate is low. Therefore, we require relatively low $\beta$ and focus more on the accurate term $\mathcal{L}_{\text{InfoNCE}}$ when noise rate is low. On the contrary, when noise rate is high, we require $\beta = 1$, which is theoretically proved to be robust in Section 4.2.

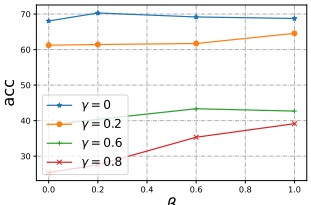 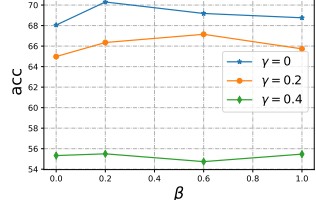 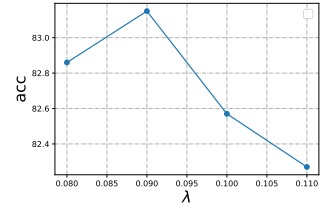

(a) Analysis of $\beta$ under symmetric label noise.    (b) Analysis of $\beta$ under asymmetric label noise.    (c) Verification of $\lambda$ choice in RINCE.

Figure 1: (a)(b) Parameter analysis of weight parameter $\beta$ in SymNCE under symmetric and asymmetric label noise. (c) Verification of theoretical choice of $\lambda$ in RINCE.

### 5.4 VERIFICATION OF OUR THEORETICAL BYPRODUCT FOR RINCE

Recall that in Section 3.5, when unifying RINCE into our theoretical framework, we could byproduct provide a theoretical optimal parameter $\lambda = 1/(K + 1)$ for RINCE. Here we conduct experiments to verify the claim. We run experiments with ResNet-18 on the CIFAR-10 dataset under 40% symmetric label noise. We vary the parameter $\lambda = \{0.08, 0.09, 0.10, 0.11\}$, and illustrate the linear probing accuracy in Figure 1(c). We show that RINCE with $\lambda = 0.09$ has the best accuracy, which coincides with our theoretical choice $\lambda = 1/(K + 1) = 1/11 \approx 0.09$.

## 6 CONCLUSION

In this paper, we proposed a unified theoretical framework for robust supervised contrastive losses against label noise. We derived a general robust condition for arbitrary contrastive losses, which further inspires us to propose the SymNCE loss, a direct robust counterpart of the widely used InfoNCE loss. As a theoretical work, our results are naturally limited by the assumptions. Nonetheless, we highlight that our theoretical analysis is a unified framework applying to multiple robust techniques such as nearest neighbor sample selection and robust losses. This framework not only provides a benchmark for assessing the resilience of contrastive losses but also holds promise as a springboard for innovative loss function designs in the future.

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

## A APPENDIX

In this appendix, we present the proofs related to Sections 3 and 4 in Appendices A.1 and A.2 respectively. We put additional training details in Appendix A.3.

### A.1 PROOFS RELATED TO SECTION 3

Note that in (Khosla et al., 2020), the supervised contrastive loss can be expressed as

$$\mathcal{L}(x, \{x_m^+\}_{m=1}^M, \{x_k^-\}_{k=1}^K; f) = \frac{1}{M} \sum_{i=1}^M \mathcal{L}_m(x, x_m^+, \{x_k^-\}_{k=1}^K; f), \tag{19}$$

and thus due to the conditional independence of $x_m^+|x$, we demonstrate the equivalence between the risks for single and multiple positive samples, i.e.

$$\begin{aligned} \mathcal{R}(\mathcal{L}; f) &= \mathbb{E}_{x, \{x_m^+\}_{m=1}^M, \{x_k^-\}_{k=1}^K} \frac{1}{M} \sum_{i=1}^M \mathcal{L}_m(x, x_m^+, \{x_k^-\}_{k=1}^K; f) \\ &= \frac{1}{M} \sum_{i=1}^M \mathbb{E}_{x, x_m^+, \{x_k^-\}_{k=1}^K} \mathcal{L}_m(x, x_m^+, \{x_k^-\}_{k=1}^K; f) \\ &= \frac{1}{M} \sum_{i=1}^M \mathcal{R}(\mathcal{L}_m; f) = \mathcal{R}(\mathcal{L}_m; f). \end{aligned} \tag{20}$$

Therefore, in the following, we analyze the case with single positive label without loss of generality.

First, in the following lemma, we derive the relationship between the clean and noisy distributions under the label corruption assumption.

**Lemma A.1.** *Under label-dependent label corruption, we have $\tilde{\pi}_i = \sum_{j=1}^C q_j(i)\pi_j$, and $\tilde{\rho}_i(x) = \left(\sum_{j=1}^C q_j(i)\rho_j(x)\pi_j\right)/\left(\sum_{j=1}^C q_j(i)\pi_j\right)$.*

*Proof of Lemma A.1.* Under label-dependent label corruption, by the law of total probability, there holds

$$\mathrm{P}(\tilde{c} = i|x) = \sum_{i=1}^C \mathrm{P}(\tilde{c} = i|c = j, x)\mathrm{P}(c = j|x) = \sum_{i=1}^C q_j(i)\mathrm{P}(c = j|x). \tag{21}$$

By taking expectation of $x$ on both sides of equation 21, we have

$$\tilde{\pi}_i = \mathrm{P}(\tilde{c} = i) = \sum_{i=1}^C q_j(i)\mathrm{P}(c = j) = \sum_{i=1}^C q_j(i)\pi_j. \tag{22}$$

On the other hand, by Bayes' theorem, there holds

$$\tilde{\rho}_i(x) = \mathrm{P}(x|\tilde{c} = i) = \frac{\mathrm{P}(\tilde{c} = i|x)\mathrm{P}(x)}{\mathrm{P}(\tilde{c} = i)}. \tag{23}$$

Then combining with equation 21 and equation 22, we have

$$\tilde{\rho}_i(x) = \frac{\sum_{i=1}^C q_j(i)\mathrm{P}(c = j|x)\mathrm{P}(x)}{\sum_{i=1}^C q_j(i)\mathrm{P}(c = j)} = \frac{\sum_{i=1}^C q_j(i)\mathrm{P}(x|c = j)\mathrm{P}(c = j)}{\sum_{i=1}^C q_j(i)\mathrm{P}(c = j)} = \frac{\sum_{i=1}^C q_j(i)\rho_j(x)\pi_j}{\sum_{i=1}^C q_j(i)\pi_j}. \tag{24}$$

$\square$

In Lemma A.2, we decompose the noisy risk for arbitrary contrastive losses without assuming any specific label types.

**Lemma A.2.** *For arbitrary contrastive loss function $\mathcal{L}(\boldsymbol{x}; f)$, the noisy contrastive risk with $M = 1$ can be decomposed into*

$$\widetilde{\mathcal{R}}(\mathcal{L}; f) = \sum_{i \in [C]} a(i)^{-1} \sum_{u \in [C]} \sum_{u^+ \in [C]} \pi_u \pi_{u^+} q_u(i) q_{u^+}(i) \mathbb{E}_{\substack{x \sim \rho_u \\ x^+ \sim \rho_{u^+}}} \mathbb{E}_{\{x_k^-\}_{k=1}^K \overset{i.i.d}{\sim} \mathrm{P}_X} \mathcal{L}(\boldsymbol{x}; f), \quad (25)$$

*where $a(i) := \sum_{u \in [C]} q_u(i) \pi_u$.*

By Lemma A.2, we see that the label noise only affects the noisy contrastive risk by changing the distribution of positive samples. The negative sample distribution remains unaffected because the negatives are uniformly drawn from the data distribution regardless of their labels. On the contrary, for positive samples selected from the noisy distribution, it is probable that a positive pair actually has different ground truth labels, and accordingly making the model overfitting to label noise.

*Proof Lemma A.2.* By definition of the noisy risk in equation 2, there holds

$$\widetilde{\mathcal{R}}(\mathcal{L}; f) = \mathbb{E}_{\tilde{c}^+, \{\tilde{c}_k^-\}_{k=1}^K \sim \tilde{\boldsymbol{\pi}}^{K+1}} \mathbb{E}_{\substack{x, x^+ \sim \tilde{\rho}_{\tilde{c}^+} \\ x_k^- \sim \tilde{\rho}_{\tilde{c}_k^-}, k \in [K]}} \mathcal{L}(x, x^+, \{x_k^-\}_{k=1}^K; f)$$

$$= \sum_{i, j_1, \dots, j_K \in [C]} \tilde{\pi}_i \tilde{\pi}_{j_1} \cdots \tilde{\pi}_{j_K} \cdot \mathbb{E}_{\substack{x, x^+ \sim \tilde{\rho}_i \\ x_k^- \sim \tilde{\rho}_{j_k}, k \in [K]}} \mathcal{L}(x, x^+, \{x_k^-\}_{k=1}^K; f)$$

$$= \sum_{i, j_1, \dots, j_K \in [C]} \sum_{m \in [C]} q_m(i) \pi_m \sum_{l_1 \in [C]} q_{l_1}(j_1) \pi_{l_1} \cdots \sum_{l_K \in [C]} q_{l_K}(j_K) \pi_{l_K}$$

$$\cdot \mathbb{E}_{\substack{x, x^+ \sim \tilde{\rho}_i \\ x_k^- \sim \tilde{\rho}_{j_k}, k \in [K]}} \mathcal{L}(x, x^+, \{x_k^-\}_{k=1}^K; f). \quad (26)$$

Denote $a(i) := \sum_{u \in [C]} q_u(i) \pi_u$ and $A(u, u^+, \boldsymbol{v}) := \mathbb{E}_{\substack{x \sim \rho_u, x^+ \sim \rho_{u^+} \\ x_k^- \sim \rho_{v_k}, k \in [K]}} \mathcal{L}(x, x^+, \{x_k^-\}_{k=1}^K; f)$. Then

by Lemma A.1, we have

$$\mathbb{E}_{\substack{x, x^+ \sim \tilde{\rho}_i \\ x_k^- \sim \tilde{\rho}_{j_k}, k \in [K]}} \mathcal{L}(x, x^+, \{x_k^-\}_{k=1}^K; f)$$

$$= \int \cdots \int \tilde{\rho}_i(x) \tilde{\rho}_i(x^+) \tilde{\rho}_{j_1}(x_1^-) \cdots \tilde{\rho}_{j_K}(x_K^-) \cdot \mathcal{L}(x, x^+, \{x_k^-\}_{k=1}^K; f) \, dx \, dx^+ \, dx_1^- \dots dx_K^-$$

$$= \int \cdots \int \frac{\sum_{u \in [C]} q_u(i) \rho_u(x) \pi_u}{\sum_{u \in [C]} q_u(i) \pi_u} \cdot \frac{\sum_{u^+ \in [C]} q_{u^+}(i) \rho_{u^+}(x^+) \pi_{u^+}}{\sum_{u^+ \in [C]} q_{u^+}(i) \pi_{u^+}} \cdot \frac{\sum_{v_1 \in [C]} q_{v_1}(j_1) \rho_{v_1}(x_1^-) \pi_{v_1}}{\sum_{v_1 \in [C]} q_{v_1}(j_1) \pi_{v_1}}$$

$$\cdots \frac{\sum_{v_K \in [C]} q_{v_K}(j_K) \rho_{v_K}(x_K^-) \pi_{v_K}}{\sum_{v_K \in [C]} q_{v_K}(j_K) \pi_{v_K}} \cdot \mathcal{L}(x, x^+, \{x_k^-\}_{k=1}^K; f) \, dx \, dx^+ \, dx_1^- \dots dx_K^-$$

$$:= (a(i)^2 a(j_1) \dots a(j_K))^{-1} \cdot \sum_{u, u^+, v_1, \dots, v_K \in [C]} \pi_u \pi_{u^+} \pi_{v_1} \cdots \pi_{v_K} \cdot q_u(i) q_{u^+}(i) q_{v_1}(j_1) \cdots q_{v_K}(j_K)$$

$$\cdot \int \cdots \int \rho_u(x) \rho_{u^+}(x^+) \rho_{v_1}(x_1^-) \cdots \rho_{v_K}(x_K^-) \cdot \mathcal{L}(x, x^+, \{x_k^-\}_{k=1}^K; f) \, dx \, dx^+ \, dx_1^- \dots dx_K^-$$

$$:= (a(i)^2 a(j_1) \dots a(j_K))^{-1} \cdot \sum_{u, u^+, v_1, \dots, v_K \in [C]} \pi_u \pi_{u^+} \pi_{v_1} \cdots \pi_{v_K} \cdot q_u(i) q_{u^+}(i) q_{v_1}(j_1) \cdots q_{v_K}(j_K)$$

$$\cdot \mathbb{E}_{\substack{x \sim \rho_u, x^+ \sim \rho_{u^+} \\ x_k^- \sim \rho_{v_k}, k \in [K]}} \mathcal{L}(x, x^+, \{x_k^-\}_{k=1}^K; f)$$

$$:= (a(i)^2 a(j_1) \dots a(j_K))^{-1} \cdot \sum_{u, u^+, v_1, \dots, v_K \in [C]} \pi_u \pi_{u^+} \pi_{v_1} \cdots \pi_{v_K}$$

$$\cdot q_u(i) q_{u^+}(i) q_{v_1}(j_1) \cdots q_{v_K}(j_K) A(u, u^+, \boldsymbol{v}). \quad (27)$$

Combining equation 26 and equation 27, we have

$$\widetilde{\mathcal{R}}(\mathcal{L}; f) = \sum_{i, j_1, \dots, j_K \in [C]} (a(i)^2 a(j_1) \dots a(j_K))^{-1}$$

$$
\cdot \sum_{m,l_1,\ldots,l_K \in [C]} q_m(i) q_{l_1}(j_1) \cdots q_{l_K}(j_K) \cdot \pi_m \pi_{l_1} \cdots \pi_{l_K}
$$

$$
\cdot \sum_{u,u^+,v_1,\ldots,v_K \in [C]} q_u(i) q_{u^+}(i) q_{v_1}(j_1) \cdots q_{v_K}(j_K) \cdot \pi_u \pi_{u^+} \pi_{v_1} \cdots \pi_{v_K} A(u, u^+, \boldsymbol{v})
$$

$$
= \sum_{i \in [C]} a(i)^{-2} \sum_{m \in [C]} \sum_{u \in [C]} \sum_{u^+ \in [C]} \pi_m \pi_u \pi_{u^+} q_m(i) q_u(i) q_{u^+}(i)
$$

$$
\cdot \sum_{j_1 \in [C]} a(j_1)^{-1} \sum_{l_1 \in [C]} \sum_{v_1 \in [C]} \pi_{l_1} \pi_{v_1} q_{l_1}(j_1) q_{v_1}(j_1) \cdots
$$

$$
\cdot \sum_{j_K \in [C]} a(j_K)^{-1} \sum_{l_K \in [C]} \sum_{v_K \in [C]} \pi_{l_K} \pi_{v_K} q_{l_K}(j_K) q_{v_K}(j_K) A(u, u^+, \boldsymbol{v}). \tag{28}
$$

For the positive term in equation 28, we have

$$
\sum_{i \in [C]} a(i)^{-2} \sum_{m \in [C]} \sum_{u \in [C]} \sum_{u^+ \in [C]} \pi_m \pi_u \pi_{u^+} q_m(i) q_u(i) q_{u^+}(i) A(u, u^+, \boldsymbol{v})
$$

$$
= \sum_{i \in [C]} a(i)^{-2} \sum_{u \in [C]} \sum_{u^+ \in [C]} \pi_u \pi_{u^+} q_u(i) q_{u^+}(i) A(u, u^+, \boldsymbol{v}) \sum_{m \in [C]} q_m(i) \pi_m
$$

$$
= \sum_{i \in [C]} a(i)^{-2} \sum_{u \in [C]} \sum_{u^+ \in [C]} \pi_u \pi_{u^+} q_u(i) q_{u^+}(i) A(u, u^+, \boldsymbol{v}) a(i)
$$

$$
= \sum_{i \in [C]} a(i)^{-1} \sum_{u \in [C]} \sum_{u^+ \in [C]} \pi_u \pi_{u^+} q_u(i) q_{u^+}(i) A(u, u^+, \boldsymbol{v}). \tag{29}
$$

For the negative terms in equation 28, we have for $k \in [K]$

$$
\sum_{j_k \in [C]} a(j_k)^{-1} \sum_{l_k \in [C]} \sum_{v_k \in [C]} \pi_{l_k} \pi_{v_k} q_{l_k}(j_k) q_{v_k}(j_k) A(u, u^+, \boldsymbol{v})
$$

$$
= \sum_{j_k \in [C]} a(j_k)^{-1} \sum_{l_k \in [C]} q_{l_k}(j_k) \pi_{l_k} \sum_{v_k \in [C]} \pi_{v_k} q_{v_k}(j_k) A(u, u^+, \boldsymbol{v})
$$

$$
= \sum_{j_k \in [C]} a(j_k)^{-1} a(j_k) \sum_{v_k \in [C]} \pi_{v_k} q_{v_k}(j_k) A(u, u^+, \boldsymbol{v})
$$

$$
= \sum_{j_k \in [C]} \sum_{v_k \in [C]} \pi_{v_k} q_{v_k}(j_k) A(u, u^+, \boldsymbol{v})
$$

$$
= \sum_{v_k \in [C]} \pi_{v_k} A(u, u^+, \boldsymbol{v}) \sum_{j_k \in [C]} q_{v_k}(j_k)
$$

$$
= \sum_{v_k \in [C]} \pi_{v_k} A(u, u^+, \boldsymbol{v}). \tag{30}
$$

Then combining equation 28, equation 29, and equation 28, we have

$$
\widetilde{\mathcal{R}}(\mathcal{L}; f) = \sum_{i \in [C]} a(i)^{-1} \sum_{u \in [C]} \sum_{u^+ \in [C]} \pi_u \pi_{u^+} q_u(i) q_{u^+}(i) \sum_{v_1,\ldots,v_K \in [C]} \pi_{v_1} \cdots \pi_{v_K} A(u, u^+, \boldsymbol{v})
$$

$$
= \sum_{i \in [C]} a(i)^{-1} \sum_{u \in [C]} \sum_{u^+ \in [C]} \pi_u \pi_{u^+} q_u(i) q_{u^+}(i) \mathbb{E}_{\substack{x \sim \rho_u, \\ x^+ \sim \rho_{u^+}}} \mathbb{E}_{\{x_k^-\}_{k=1}^K \overset{\text{i.i.d.}}{\sim} \mathrm{P}_X} \mathcal{L}(\boldsymbol{x}; f). \tag{31}
$$

$\square$

### A.1.1 PROOFS FOR SYMMETRIC LABEL NOISE

Then by Assumptions 3.1, we prove Theorem 3.2.

*Proof of Theorem 3.2.* Under Assumption 3.1, $q_i(i) = 1 - \gamma$ and $q_u(i) = \gamma/(C-1)$ for $u \neq i$. Denote $B(u, u^+) := \sum_{v_1,\dots,v_K \in [C]} \pi_{v_1} \cdots \pi_{v_K} A(u, u^+, \boldsymbol{v})$. By Lemma A.2, we have

$$
\widetilde{\mathcal{R}}(\mathcal{L}; f) = \sum_{i \in [C]} a(i)^{-1} \sum_{u \in [C]} \sum_{u^+ \in [C]} \pi_u \pi_{u^+} q_u(i) q_{u^+}(i) B(u, u^+)
$$

$$
= \sum_{i \in [C]} a(i)^{-1}(1-\gamma)^2 \pi_i^2 B(i,i) + \sum_{i \in [C]} a(i)^{-1} \frac{\gamma(1-\gamma)}{C-1} \pi_i \sum_{u^+ \neq i} \pi_{u^+} B(i, u^+)
$$

$$
+ \sum_{i \in [C]} a(i)^{-1} \frac{\gamma(1-\gamma)}{C-1} \pi_i \sum_{u \neq i} \pi_u B(u, i) + \sum_{i \in [C]} a(i)^{-1} \frac{\gamma^2}{(C-1)^2} \sum_{u \neq i} \sum_{u^+ \neq i} \pi_u \pi_{u^+} B(u, u^+)
$$

$$
= \sum_{i \in [C]} a(i)^{-1}\Big((1-\gamma)^2 - \frac{\gamma^2}{(C-1)^2}\Big) \pi_i^2 B(i,i)
$$

$$
+ \sum_{i \in [C]} a(i)^{-1}\Big(\frac{\gamma(1-\gamma)}{C-1} - \frac{\gamma^2}{(C-1)^2}\Big) \pi_i \sum_{u^+ \neq i} \pi_{u^+} B(i, u^+)
$$

$$
+ \sum_{i \in [C]} a(i)^{-1}\Big(\frac{\gamma(1-\gamma)}{C-1} - \frac{\gamma^2}{(C-1)^2}\Big) \pi_i \sum_{u \neq i} \pi_u B(u, i)
$$

$$
+ \sum_{i \in [C]} a(i)^{-1} \frac{\gamma^2}{(C-1)^2} \sum_{u \in [C]} \sum_{u^+ \in [C]} \pi_u \pi_{u^+} B(u, u^+)
$$

$$
= \sum_{i \in [C]} a(i)^{-1}\Big((1-\gamma)^2 - \frac{2\gamma}{C-1}\Big(1 - \frac{C}{C-1}\gamma\Big) - \frac{\gamma^2}{(C-1)^2}\Big) \pi_i^2 B(i,i)
$$

$$
+ \sum_{i \in [C]} a(i)^{-1} \frac{\gamma}{C-1}\Big(1 - \frac{C}{C-1}\gamma\Big) \pi_i \sum_{u^+ \in [C]} \pi_{u^+} B(i, u^+)
$$

$$
+ \sum_{i \in [C]} a(i)^{-1} \frac{\gamma}{C-1}\Big(1 - \frac{C}{C-1}\gamma\Big) \pi_i \sum_{u \in [C]} \pi_u B(u, i)
$$

$$
+ \sum_{i \in [C]} a(i)^{-1} \frac{\gamma^2}{(C-1)^2} \sum_{u \in [C]} \sum_{u^+ \in [C]} \pi_u \pi_{u^+} B(u, u^+)
$$

$$
= \sum_{u \in [C]} a(u)^{-1}\Big(1 - \frac{C}{C-1}\gamma\Big)^2 \pi_u^2 B(u, u)
$$

$$
+ \sum_{u \in [C]} a(u)^{-1} \frac{\gamma}{C-1}\Big(1 - \frac{C}{C-1}\gamma\Big) \pi_u \sum_{u^+ \in [C]} \pi_{u^+} B(u, u^+)
$$

$$
+ \sum_{u^+ \in [C]} a(u^+)^{-1} \frac{\gamma}{C-1}\Big(1 - \frac{C}{C-1}\gamma\Big) \pi_{u^+} \sum_{u \in [C]} \pi_u B(u, u^+)
$$

$$
+ \sum_{i \in [C]} a(i)^{-1} \frac{\gamma^2}{(C-1)^2} \sum_{u \in [C]} \sum_{u^+ \in [C]} \pi_u \pi_{u^+} B(u, u^+). \tag{32}
$$

When the input data is class balanced, i.e. $\pi_i = \frac{1}{C}$ for $i \in [C]$, we have for $i \in [C]$,

$$
a(i) = \sum_{u \in [C]} q_u(i) \pi_u = \frac{1}{C} \sum_{u \in [C]} q_u(i) = \frac{1}{C}. \tag{33}
$$

Then we have

$$
\widetilde{\mathcal{R}}(\mathcal{L}; f)
$$

$$
= \Big(1 - \frac{C}{C-1}\gamma\Big)^2 \sum_{u \in [C]} \pi_u B(u, u) + \frac{C\gamma}{C-1}\Big(1 - \frac{C}{C-1}\gamma\Big) \sum_{u \in [C]} \pi_u \sum_{u^+ \in [C]} \pi_{u^+} B(u, u^+)
$$

$$+ \frac{C\gamma}{C-1}\Big(1 - \frac{C}{C-1}\gamma\Big) \sum_{u^+ \in [C]} \pi_{u^+} \sum_{u \in [C]} \pi_u B(u, u^+) + \frac{C^2\gamma^2}{(C-1)^2} \sum_{u \in [C]} \sum_{u^+ \in [C]} \pi_u \pi_{u^+} B(u, u^+)$$

$$= \Big(1 - \frac{C}{C-1}\gamma\Big)^2 \sum_{u \in [C]} \pi_u B(u, u) + \frac{C\gamma}{C-1}\Big(2 - \frac{C}{C-1}\gamma\Big) \sum_{u, u^+ \in [C]} \pi_u \pi_{u^+} B(u, u^+) \qquad (34)$$

Note that

$$\sum_{u \in [C]} \pi_u B(u, u) = \sum_{u \in [C]} \pi_u \sum_{v_1, \dots, v_K \in [C]} \pi_{v_1} \cdots \pi_{v_K} \mathbb{E}_{\substack{x \sim \rho_u, x^+ \sim \rho_u \\ x_k^- \sim \rho_{v_k}, k \in [K]}} \mathcal{L}(x, x^+, \{x_k^-\}_{k=1}^K; f)$$

$$= \mathbb{E}_{c^+, \{c_k^-\}_{k=1}^K \sim \boldsymbol{\pi}^{K+1}} \mathbb{E}_{\substack{x \sim \rho_u, x^+ \sim \rho_u \\ x_k^- \sim \rho_{v_k}, k \in [K]}} \mathcal{L}(x, x^+, \{x_k^-\}_{k=1}^K; f)$$

$$= \mathcal{R}(\mathcal{L}; f), \qquad (35)$$

and that

$$\sum_{u, u^+ \in [C]} \pi_u \pi_{u^+} B(u, u^+)$$

$$= \sum_{u, u^+ \in [C]} \pi_u \pi_{u^+} \sum_{v_1, \dots, v_K \in [C]} \pi_{v_1} \cdots \pi_{v_K} \mathbb{E}_{\substack{x \sim \rho_u, x^+ \sim \rho_{u^+} \\ x_k^- \sim \rho_{v_k}, k \in [K]}} \mathcal{L}(x, x^+, \{x_k^-\}_{k=1}^K; f)$$

$$= \mathbb{E}_{c, c^+, \{c_k^-\}_{k=1}^K \sim \boldsymbol{\pi}^{K+2}} \mathbb{E}_{\substack{x \sim \rho_u, x^+ \sim \rho_{u^+} \\ x_k^- \sim \rho_{v_k}, k \in [K]}} \mathcal{L}(x, x^+, \{x_k^-\}_{k=1}^K; f)$$

$$:= \Delta\mathcal{R}(\mathcal{L}; f). \qquad (36)$$

Thus we have

$$\widetilde{\mathcal{R}}(\mathcal{L}; f) = \Big(1 - \frac{C}{C-1}\gamma\Big)^2 \mathcal{R}(\mathcal{L}; f) + \frac{C\gamma}{C-1}\Big(2 - \frac{C}{C-1}\gamma\Big)\Delta\mathcal{R}(\mathcal{L}; f). \qquad (37)$$

$\square$

*Proof of Corollary 3.3.* For symmetric label noise, by Theorem 3.2 and that $\Delta\mathcal{R}(\mathcal{L}; f) = A$, we have

$$\widetilde{\mathcal{R}}(\mathcal{L}; f) = \Big(1 - \frac{C}{C-1}\gamma\Big)^2 \mathcal{R}(\mathcal{L}; f) + \frac{C\gamma}{C-1}\Big(2 - \frac{C}{C-1}\gamma\Big)A. \qquad (38)$$

Suppose $f^*$ is the minimizer of $\widetilde{\mathcal{R}}(\mathcal{L}; f)$, i.e. for all $f$

$$\widetilde{\mathcal{R}}(\mathcal{L}; f^*) \le \widetilde{\mathcal{R}}(\mathcal{L}; f). \qquad (39)$$

Then if $\gamma \le (C-1)/C$, we have

$$\mathcal{R}(\mathcal{L}; f^*) \le \mathcal{R}(\mathcal{L}; f), \qquad (40)$$

that is, $f^*$ is also the minimizer of $\mathcal{R}(\mathcal{L}; f)$. $\square$

### A.1.2 PROOFS FOR ASYMMETRIC LABEL NOISE

Next, we show the results under asymmetric label noise.

**Assumption A.3** (Asymmetric label noise)**.** *For noise rates $\gamma_i \in (0, 1)$, $i \in [C]$, we assume that there holds $q_i(i) = 1 - \gamma_i$ and $q_j(i) = \gamma_{ij} \ge 0$ for all $j \ne i$.*

**Theorem A.4.** *Assume that the input data is class balanced, i.e. $\pi_i = \tilde{\pi}_i = 1/C$ for $i \in [C]$. Then under Assumption A.3, if we have $\sum_{i \in [C]} \gamma_{iu}^2 = c_1(\boldsymbol{\gamma})$ and $\sum_{i \in [C]} \gamma_{iu}\gamma_{iu^+} = c_2(\boldsymbol{\gamma})$ for all $u \ne u^+ \in [C]$, then for an arbitrary contrastive loss $\mathcal{L}(\boldsymbol{x}; f)$, where $c_1(\boldsymbol{\gamma})$ and $c_2(\boldsymbol{\gamma})$ are constants related to $\boldsymbol{\gamma} := (\gamma_{ij})_{i,j \in [C]}$, there holds*

$$\widetilde{\mathcal{R}}(\mathcal{L}; f) = \big(c_1(\boldsymbol{\gamma}) - c_2(\boldsymbol{\gamma})\big)\mathcal{R}(\mathcal{L}; f) + C \cdot c_2(\boldsymbol{\gamma})\Delta\mathcal{R}(\mathcal{L}; f), \qquad (41)$$

*where*

$$\Delta\mathcal{R}(\mathcal{L}; f) := \mathbb{E}_{\substack{c^+, \{c_m^+\}_{m=1}^M \sim \boldsymbol{\pi}^{M+1} \\ \{c_k^-\}_{k=1}^K \sim \boldsymbol{\pi}^K}} \mathbb{E}_{\substack{x \sim \rho_{c^+}, x_m^+ \sim \rho_{c_m^+}, m \in [M] \\ x_k^- \sim \rho_{c_k^-}, k \in [K]}} \mathcal{L}(\boldsymbol{x}; f). \qquad (42)$$

*Proof of Theorem A.4.*

$$
\begin{aligned}
\widetilde{\mathcal{R}}(\mathcal{L}; f) &= \sum_{i\in[C]} a(i)^{-1} \sum_{u\in[C]} \sum_{u^+\in[C]} \pi_u \pi_{u^+} q_u(i) q_{u^+}(i) B(u, u^+) \\
&= \sum_{i\in[C]} a(i)^{-1}(1-\gamma_i)^2 \pi_i^2 B(i,i) + \sum_{i\in[C]} a(i)^{-1} \sum_{u\neq i} \gamma_{iu}(1-\gamma_i)\pi_u \pi_i B(u,i) \\
&\quad + \sum_{i\in[C]} a(i)^{-1} \sum_{u^+\neq i} \gamma_{iu^+}(1-\gamma_i)\pi_{u^+}\pi_i B(u, u^+) \\
&\quad + \sum_{i\in[C]} a(i)^{-1} \sum_{u\neq i}\sum_{u^+\neq i} \pi_u \pi_{u^+} \gamma_{iu}\gamma_{iu^+} B(i, u^+) \\
&= \sum_{u\in[C]} a(u)^{-1}(1-\gamma_u)^2 \pi_u^2 B(u,u) + \sum_{u^+\in[C]} a(u^+)^{-1} \sum_{u\neq u^+} \gamma_{u^+u}(1-\gamma_{u^+})\pi_u \pi_{u^+} B(u, u^+) \\
&\quad + \sum_{u\in[C]} a(u)^{-1} \sum_{u^+\neq u} \gamma_{uu^+}(1-\gamma_u)\pi_{u^+}\pi_u B(u, u^+) \\
&\quad + \sum_{i\in[C]} a(i)^{-1} \sum_{u\neq i} \pi_u^2 \gamma_{iu}^2 B(u,u) + \sum_{i\in[C]} a(i)^{-1} \sum_{u\neq i}\sum_{u^+\neq u, i} \pi_u \pi_{u^+} \gamma_{iu}\gamma_{iu^+} B(u, u^+) \\
&= \sum_{u\in[C]} a(u)^{-1}(1-\gamma_u)^2 \pi_u^2 B(u,u) + \sum_{u^+\in[C]}\sum_{u\neq u^+} a(u^+)^{-1}\gamma_{u^+u}(1-\gamma_{u^+})\pi_u\pi_{u^+} B(u, u^+) \\
&\quad + \sum_{u\in[C]}\sum_{u^+\neq u} a(u)^{-1}\gamma_{uu^+}(1-\gamma_u)\pi_u\pi_{u^+} B(u, u^+) \\
&\quad + \sum_{u\in[C]}\sum_{i\neq u} a(i)^{-1}\gamma_{iu}^2 \pi_u^2 B(u,u) + \sum_{u\in[C]}\sum_{u^+\neq u}\sum_{i\neq u, u^+} a(i)^{-1}\gamma_{iu}\gamma_{iu^+}\pi_u\pi_{u^+} B(u, u^+).
\end{aligned}
\tag{43}
$$

Because $a(i) = \tilde{\pi}_i = 1/C$ for all $i \in [C]$, we have

$$
\begin{aligned}
\widetilde{\mathcal{R}}(\mathcal{L}; f) &= C\sum_{u\in[C]}(1-\gamma_u)^2 \pi_u^2 B(u,u) + C\sum_{u^+\in[C]}\sum_{u\neq u^+}\gamma_{u^+u}(1-\gamma_{u^+})\pi_u\pi_{u^+} B(u, u^+) \\
&\quad + C\sum_{u\in[C]}\sum_{u^+\neq u}\gamma_{uu^+}(1-\gamma_u)\pi_u\pi_{u^+} B(u, u^+) \\
&\quad + C\sum_{u\in[C]}\sum_{i\neq u}\gamma_{iu}^2 \pi_u^2 B(u,u) + C\sum_{u\in[C]}\sum_{u^+\neq u}\sum_{i\neq u,u^+}\gamma_{iu}\gamma_{iu^+}\pi_u\pi_{u^+} B(u, u^+) \\
&= C\sum_{u\in[C]}\Big[(1-\gamma_u)^2 + \sum_{i\neq u}\gamma_{iu}^2\Big]\pi_u^2 B(u,u) \\
&\quad + C\sum_{u\in[C]}\sum_{u^+\neq u}\Big[\sum_{i\neq u, u^+}\gamma_{iu}\gamma_{iu^+} + \gamma_{u^+u}(1-\gamma_{u^+}) + \gamma_{uu^+}(1-\gamma_u)\Big]\pi_u\pi_{u^+} B(u, u^+) \\
&= C\sum_{u\in[C]}\Big(\sum_{i\in[C]}\gamma_{iu}^2\Big)\pi_u^2 B(u,u) + C\sum_{u\in[C]}\sum_{u^+\neq u}\Big(\sum_{i\in[C]}\gamma_{iu}\gamma_{iu^+}\Big)\pi_u\pi_{u^+} B(u, u^+). \quad (44)
\end{aligned}
$$

By assumption, $\sum_{i\in[C]}\gamma_{iu}^2 = c_1(\boldsymbol{\gamma})$ and $\sum_{i\in[C]}\gamma_{iu}\gamma_{iu^+} = c_2(\boldsymbol{\gamma})$ for all $u \neq u^+ \in [C]$. Thus we have

$$
\begin{aligned}
\widetilde{\mathcal{R}}(\mathcal{L}; f) &= C\cdot c_1(\boldsymbol{\gamma})\sum_{u\in[C]}\pi_u^2 B(u,u) + C\cdot c_2(\boldsymbol{\gamma})\sum_{u\in[C]}\sum_{u^+\neq u}\pi_u\pi_{u^+} B(u, u^+) \\
&= C\big(c_1(\boldsymbol{\gamma}) - c_2(\boldsymbol{\gamma})\big)\sum_{u\in[C]}\pi_u^2 B(u,u) + C\cdot c_2(\boldsymbol{\gamma})\sum_{u, u^+\in[C]}\pi_u\pi_{u^+} B(u, u^+) \\
&= \big(c_1(\boldsymbol{\gamma}) - c_2(\boldsymbol{\gamma})\big)\mathcal{R}(\mathcal{L}; f) + C\cdot c_2(\boldsymbol{\gamma})\Delta\mathcal{R}(\mathcal{L}; f).
\end{aligned}
\tag{45}
$$

$\square$

In Corollary 3.3, we give the general condition for an arbitrary contrastive loss function to be noise tolerant under asymmetric label noise. Comparing with Ghosh et al. (2017), our theoretical framework requires the same noise rate $1 - \gamma_u > \gamma_{ui}$ for all $i \neq u$, and $u, i \in [C]$.

**Theorem A.5.** *Assume that the input data is class balanced, and there exists a constant $A \in \mathbb{R}$ such that $\Delta \mathcal{R}(\mathcal{L}; f) = A$. Then under Assumption A.3, contrastive loss $\mathcal{L}$ is noise tolerant if $1 - \gamma_u > \gamma_{ui}$ for all $i \neq u$, $u, i \in [C]$.*

*Proof of Theorem A.5.* When $\Delta \mathcal{R}(\mathcal{L}; f) = A$, we have

$$\widetilde{\mathcal{R}}(\mathcal{L}; f) = \big(c_1(\boldsymbol{\gamma}) - c_2(\boldsymbol{\gamma})\big)\mathcal{R}(\mathcal{L}; f) + CA \cdot c_2(\boldsymbol{\gamma}). \tag{46}$$

Then we calculate $c_1(\boldsymbol{\gamma}) - c_2(\boldsymbol{\gamma})$.

$$\begin{aligned}
c_1(\boldsymbol{\gamma}) - c_2(\boldsymbol{\gamma}) &= \sum_{i \in [C]} \gamma_{iu}^2 - \sum_{i \in [C]} \gamma_{iu}\gamma_{iu^+} \\
&= \frac{1}{2} \sum_{i \in [C]} \big(\gamma_{iu}^2 + \gamma_{iu^+}^2 - 2\gamma_{iu}\gamma_{iu^+}\big) \\
&= \frac{1}{2} \sum_{i \in [C]} (\gamma_{iu} - \gamma_{iu^+})^2 \\
&= \frac{1}{2}\Big[ \sum_{i \neq u, u^+} (\gamma_{iu} - \gamma_{iu^+})^2 + (1 - \gamma_u - \gamma_{uu^+})^2 + (1 - \gamma_{u^+} - \gamma_{u^+u})^2 \Big]. \tag{47}
\end{aligned}$$

If $1 - \gamma_u > \gamma_{ui}$ for all $i \neq u$, $u, i \in [C]$, then $c_1(\boldsymbol{\gamma}) - c_2(\boldsymbol{\gamma}) > 0$. Suppose $f^*$ is the minimizer of $\widetilde{\mathcal{R}}(\mathcal{L}; f)$, and thus we have when $1 - \gamma_u > \gamma_{ui}$,

$$\mathcal{R}(\mathcal{L}; f) - \mathcal{R}(\mathcal{L}; f^*) = \frac{1}{\big(c_1(\boldsymbol{\gamma}) - c_2(\boldsymbol{\gamma})\big)}\Big[\widetilde{\mathcal{R}}(\mathcal{L}; f^*) - \widetilde{\mathcal{R}}(\mathcal{L}; f)\Big] < 0, \tag{48}$$

that is, $f^*$ is also the minimizer of $\mathcal{R}(\mathcal{L}; f)$. $\qquad\square$

## A.2 PROOFS RELATED TO SECTION 4

For completeness, we first prove the limit form of InfoNCE shown in equation 6, following Wang & Isola (2020).

$$\begin{aligned}
&\lim_{M,K \to \infty} \Delta \mathcal{R}(\mathcal{L}_{\text{InfoNCE}}; f) - \log K \\
&= \lim_{M,K \to \infty} \mathbb{E}_{x, x_m^+, x_k^- \overset{\text{i.i.d}}{\sim} \mathrm{P}_X} \frac{1}{M} \sum_{m \in [M]} -\log \frac{e^{f(x)^\top f(x_m^+)}}{e^{f(x)^\top f(x_m^+)} + \sum_{k \in [K]} e^{f(x)^\top f(x_k^-)}} - \log K \\
&= \mathbb{E}_{x, x_m^+, x_k^- \overset{\text{i.i.d}}{\sim} \mathrm{P}_X} -f(x)^\top f(x_m^+) + \lim_{K \to \infty} \log \Big(\frac{1}{K} e^{f(x)^\top f(x_m^+)} + \frac{1}{K} \sum_{k \in [K]} e^{f(x)^\top f(x_k^-)}\Big) \\
&= \mathbb{E}_{x, x^+ \overset{\text{i.i.d}}{\sim} \mathrm{P}_X} -f(x)^\top f(x^+) + \mathbb{E}_{x \sim \mathrm{P}_X} \log \big(\mathbb{E}_{x^- \sim \mathrm{P}_X} e^{f(x)^\top f(x^-)}\big). \tag{49}
\end{aligned}$$

*Proof of Proposition 4.1.*

$$\begin{aligned}
&\lim_{M,K \to \infty} \Delta \mathcal{R}(\mathcal{L}_{\text{RevNCE}}; f) \\
&= \lim_{K,M \to \infty} \mathbb{E}_{x, x_m^+, x_k^- \overset{\text{i.i.d}}{\sim} \mathrm{P}_X} \frac{1}{K} \sum_{k \in [K]} -\log \frac{\frac{1}{M} \sum_{m \in [M]} e^{f(x)^\top f(x_m^+)}}{e^{f(x)^\top f(x_k^-)}} \\
&= \mathbb{E}_{x, x_m^+, x_k^- \overset{\text{i.i.d}}{\sim} \mathrm{P}_X} \lim_{K,M \to \infty} \frac{1}{K} \sum_{k \in [K]} -\log \Big(\frac{1}{M} \sum_{m \in [M]} e^{f(x)^\top f(x_m^+)}\Big) + \frac{1}{K} \sum_{k \in [K]} f(x)^\top f(x_k^-)
\end{aligned}$$

$$
\begin{aligned}
&= \mathbb{E}_{x \sim \mathrm{P}_X} - \log\left(\mathbb{E}_{x^+ \sim \mathrm{P}_X} e^{f(x)^\top f(x^+)}\right) + \mathbb{E}_{x,x^- \overset{\text{i.i.d}}{\sim} \mathrm{P}_X} f(x)^\top f(x_k^-) \\
&= \lim_{M,K \to \infty} -\Delta\mathcal{R}(\mathcal{L}_{\text{InfoNCE}}; f) + \log K.
\end{aligned}
\tag{50}
$$

$\square$

## A.3 TRAINING DETAILS

**Training details on CIFAR datasets.** We run all experiments on an NVIDIA GeForce RTX 3090 24G GPU. We adopt a ResNet-18 as the backbone for all methods, and use the SGD optimizer with momentum 0.9. The batch size is set to be 512. For our SymNCE, we set learning rate as 0.1 for CIFAR-10 and 0.01 for CIFAR-100 without decay. Temperature is 0.5 for CIFAR-10 and 0.07 for CIFAR-100. Weight decay is set to be $10^{-4}$. The weight parameter $\beta$ is selected in $\{0.2, 0.6, 1.0\}$ through validation. For robust start, we warm up with the SupCon loss in the early stage of training. For compared methods, the parameters are set referring to their original papers. For all supervised contrastive losses, we run 300 epochs for training the representations, which are then evaluated through linear probing on the noisy dataset with CE loss for 30 epochs. The data augmentations are random crop and resize (with random flip), color distortion and color dropping. For supervised classification losses, we train all losses for 300 epochs and report the test accuracy.

For SupCon, we set learning rate as 0.5 for CIFAR-10 and 0.01 for CIFAR-100 without decay. Temperature is 0.1 for CIFAR-10 and 0.07 for CIFAR-100. Weight decay is set to be $10^{-4}$. For RINCE, we set temperature 0.1, $\lambda = 0.01$ and $q = 0.5$. For RINCE, SCE, AP, CE, and MAE, the learning rate is set to be 0.01 without decay. We set $q = 0.7$ in GCE, $\alpha = 0.1$, $\beta = 1$ for SCE. For APL (NCE+MAE), we set $\alpha = \beta = 1$ for CIFAR-10 and $\alpha = 10$, $\beta = 0.1$ for CIFAR-100.

**Training details on Clothing1M dataset.** We adopt an ImageNet pre-trained ResNet-18 as our backbone. We use the SGD optimizer with momentum 0.9. The batchsize is set to be 32. For our method, we set learning rate as 0.002. The temperature is 0.07 and weigth decay is set to be $10^{-4}$. For robust start, we warm up with the SupCon loss for the early 20 epoches. For all supervised contrastive losses, we run 80 epoches for representation learning,which are then evaluated through the linear probing on the training dataset with CE loss for 30 epoches. For compared method, we all set the training epoches as 80 for a fair comparison.

