# OpenReview forum: "A Unified Framework of Theoretically Robust Contrastive Loss against Label Noise"
_ICLR.cc/2024/Conference — Submitted to ICLR 2024_

### Official Review · Reviewer_73JG · 2023-10-28

**Soundness:** 3 good
**Presentation:** 3 good
**Contribution:** 3 good
**Rating:** 6
**Confidence:** 3

**Summary:**

This paper analyzes robust supervised contrastive learning losses under noisy labels. It proposes to analyze from a theoretical perspective to formally define the classification risks under both clean and noisy labels. It then claims that the difference term being a constant is the condition for a contrastive loss to be robust. The condition is used provide better insights for existing methods and propose a modification to InfoNCE.

**Strengths:**

- The paper focuses on a practially important problem and propose a theoretically grounded solution.
- The proposed formulation is highly extendable to other losses. It is of high significance.

**Weaknesses:**

- The assumption of class-balanced distribution and symmetric labels noise might be strong.
- Some parts of the presentation can be improved.
  - For example, in Section 3.4.1, it would be nice to add intuitive explanations to each term for readers to understand the derivation better.

**Questions:**

- Dicussion after Corollary 3.3 is a bit confusing. Does Ghosh et al. 2017 also need the assumption of class-balanced distribution? What is the conclusion of the discussion, is the proposed framework better or both have there merits?
- Experiments also show promising results on asymmetric noises. Are there some theoretical insights or difficulties for the proposed framework to work with asymmetric noises?

---

### Official Review · Reviewer_qnEx · 2023-11-03

**Soundness:** 3 good
**Presentation:** 2 fair
**Contribution:** 2 fair
**Rating:** 5
**Confidence:** 4

**Summary:**

The paper addresses the challenge of learning from noisy labels in machine learning, focusing on the use of supervised contrastive learning. It proposes a unified theoretical framework for constructing robust supervised contrastive losses, presenting a general condition for assessing the robustness of these losses against label noise. The authors develop Symmetric InfoNCE (SymNCE), a robust version of the InfoNCE loss, and their extensive experiments show that SymNCE outperforms existing methods in handling label noise on benchmark datasets.

**Strengths:**

1. The authors proposed a unified framework for analyzing the robust supervised contrastive loss.
2. The authors proposed a noise-robust contrastive loss function and proved its optimality under some assumptions.
3. Solid theoretical analyses.

**Weaknesses:**

1. Corollary 3.3 assumes a balanced input (clean) data distribution. However, in practice, it is not clear if the clean data distribution is balanced or not without ground-truth labels.
2. It is okay if the assumption is just for a clear presentation. In other words, the authors should show the effectiveness of the proposed loss function when the input data distribution is imbalanced, which is more practical.
3. The synthetic label noise is too simple. Instance-dependent label noise [1,2] on CIFAR should be considered. Besides, real-world human annotations should also be considered [3,4].
4. It is unfair to use only 1.4k samples in Clothing1M since many baselines are originally designed on the whole 1M noisy training dataset.


[1] Learning with Instance-Dependent Label Noise: A Sample Sieve Approach. ICLR 2021.

[2] Part-dependent label noise: Towards instance-dependent label noise. NeurIPS 2020.

[3] Human uncertainty makes classification more robust. CVPR 2019.

[4] Learning with Noisy Labels Revisited: A Study Using Real-World Human Annotations. ICLR 2022.

**Questions:**

Please see the weakness above.

---

### Official Review · Reviewer_9C43 · 2023-11-03

**Soundness:** 2 fair
**Presentation:** 2 fair
**Contribution:** 2 fair
**Rating:** 5
**Confidence:** 3

**Summary:**

The authors propose a framework for studying the theoretical robustness of contrastive losses, similar to Ghosh et al.’s framework for supervised losses [1]. As in Ghosh et al.'s framework, the authors break down the risk with noisy labels into two component: a risk for the true labels and an additional risk. Again, as in Ghosh et al.'s framework, if this additional risk is constant for any function f, the loss is noise tolerant.

The authors study if this additional risk is a constant for InfoNCE, InfoNCE-NN, and RINCE. Their findings indicate that InfoNCE is not noise-tolerant, whereas the other two are. Inspired by the symmetric cross entropy loss for supervised losses [2], the authors propose SymNCE which is InfoNCE with an additional reverse InfoNCE loss, which they show is noise-tolerant.

The authors perform experiments to compare their proposed loss with some theoretically robust losses based on Ghosh et al., as well as two contrastive loss functions.

**References.**

[1] Ghosh A, Kumar H, Sastry PS. Robust loss functions under label noise for deep neural networks.

[2] Wang Y, Ma X, Chen Z, Luo Y, Yi J, Bailey J. Symmetric cross entropy for robust learning with noisy labels

**Strengths:**

* _Originality_: Investigating the theoretical robustness of contrastive loss functions is not entirely novel, as RINCE was the first work extending Ghosh et al.’s theory for doing so. However, what sets this work apart is a theory applicable to multiple loss functions.

* _Quality_: All baselines were implemented in the same code base, which makes it possible to do fair comparisons. Although the theory is only for synthetic noise, I believe it was important that the empirical robustness of the methods were studied under realistic/natural noise conditions, i.e., Clothing1M.

* _Clarity_: The paper has a well-organized structure, with an introduction that clearly states the contributions of the work.

* _Significance_: Establishing a proper theoretical framework for studying the robustness of loss functions, be they supervised or contrastive or other, is an important goal. This work represents an incremental step in the direction of achieving this goal.

**Weaknesses:**

**Unclear Theory.**

I found several parts of the theory to be unclear, e.g., some steps in a proof, what previous work the theory is based on, and justifications for the robustness of InfoNCE-NN, etc. These concerns are outlined in the Questions section below.

**Problematic Empirical Comparisons.**

In the list of contributions, it is claimed that the proposed SymNCE is comparable or outperforms existing state-of-the-art robust loss functions. However, the empirical comparisons presented in this study raise several concerns, which need to be addressed:

  * _State-of-the-art robust loss functions_: The claim that SymNCE outperforms state-of-the-art robust loss functions is problematic, as the baselines are outdated. For example, Ghosh et al.’s theory has been extended, resulting in an entire family of “asymmetric” noise robust loss functions [3]. Furthermore, the GJS loss [3] builds on Ghosh et al’s theory and incorporates consistency regularization (instead of contrastive learning) for improved robustness.
If the authors want to claim state-of-the-art performance, then I believe these should be included in the comparisons.

  * _Unreliable comparisons_: Claims about performance improvements are unreliable due to absence of reporting mean and standard deviation from multiple training runs with different random seeds. This is important to assess the stability and statistical significance of the results.


  * _Unfair comparisons_: In Appendix A.3, the training details are described.
	* Warmup: The inclusion of warmup using SupCon for SymNCE, while not being applied to other baselines, creates an unfair comparison. It's important to maintain consistency in training approaches across methods.
	* Varying method-specific hyperparameters: SymNCE is given more flexibility in changing its hyperparameters for different noise rates, noise types, and datasets compared to baselines, resulting in an unfair advantage. For baselines, it is said that “For compared methods, the parameters are set referring to their original papers.” where some methods like RINCE, SCE, and GCE have to use the same HPs for all noise rates and datasets. However, for SymNCE: “The weight parameter $\beta$ is selected in \{0.2, 0.6, 1.0 \} through validation.” From this description, it is unclear if this $\beta$ was selected per dataset or not. Comparing the results in Figure 1 (a, b) with the results in Table 1, it seems $\beta$ is selected per noise rate, type, and dataset. For example, for symmetric noise on CIFAR-100, it seems $\beta$ is 0.2, 1.0, 0.6, 1.0 for the respective noise rates $\gamma$ in 0.0, 0.2, 0.6, 0.8.  This is not the case for the baselines, that use a fixed set of HPs for each dataset, which makes the comparison unfair. Furthermore, SupCon, SymNCE, and RINCE all depend on InfoNCE that has a temperature parameter. In the experiments with RINCE, the temperature is fixed for all datasets, but for SupCon (warmup of SymNCE) and SymNCE the temperature is changed for the different datasets, which is again unfair.
	* Varying learning rate: The learning rate is the same for all baselines (except SupCon) on both CIFAR-10 and CIFAR-100, but for SupCon (warmup of SymNCE) and SymNCE it is changed per dataset. Again, this is unfair.
	* Batch size for contrastive losses: In addition to having fixed learning rate, and fixed method-specific parameters, I am concerned if the poor performance of the supervised loss functions is also due to the larger than usual batch size (512). Larger batch sizes have been shown to help in contrastive learning, and I therefore expect it to be good for SupCon, RINCE and SymNCE, but not necessarily for the other baselines, which could be unfair.


For valid and fair empirical comparisons, it is important to maintain consistent training protocols and hyperparameter searches across all methods under evaluation. Additionally, reporting results from multiple training runs with different random seeds will enhance the reliability of the findings.

**Verification of the theory for the choice of $\lambda$ in RINCE.**

As there is little variation in the y-axis in Figure 1c, I again believe it is crucial to report results from multiple training runs with different random seeds to draw any valid conclusions.

**Missing related work.**

The list of robust loss functions is not comprehensive, e.g., the following relevant works are missing: the asymmetric loss functions [2], and robust losses based on information theory: i) Jensen-Shannon divergences [3], ii) f-divergences [4], and iii) Bregman divergences [5].

**References.**

[1] Ghosh A, Kumar H, Sastry PS. Robust loss functions under label noise for deep neural networks.

[2] Zhou X, Liu X, Jiang J, Gao X, Ji X. Asymmetric loss functions for learning with noisy labels.

[3] Englesson E, Azizpour H. Generalized jensen-shannon divergence loss for learning with noisy labels.

[4] Wei J, Liu Y. When optimizing $ f $-divergence is robust with label noise.

[5] Amid E, Warmuth MK, Anil R, Koren T. Robust bi-tempered logistic loss based on bregman divergences.

**Questions:**

**Experimental Clarifications.**

* Are all baselines trained with the same data augmentation strategy?
* In the first sentence in Section 5.3, it is claimed that $\beta$ is the only parameter of SymNCE. What about the temperature $\tau$ in Equations 17 and 18 which is changed for different datasets?
* For your method, how important is warmup with SupCon? Could you provide results without warmup?
* How many positive and negative samples are used in the experiments for the contrastive losses?
* Why do you believe the results are so poor on Clothing1M, e.g., 53.48 for CE? For example, some results with a ResNet 50 and the CE loss from the literature are: 68.80 [1], 69.21 [2], 69.10 [3], and Yi et al. [4] reports a result of 70.88 with CE and ResNet 18.

**Theoretical Clarifications.**
* Is there one or more missing steps going from Equation 39 to 40 in the proof of Corollary 3.3? See for example the proof of Theorem 1 in Ghosh et al. [5]. I think this step is crucial, as it shows that the constant has to be independent of f. This is for example important in Corollary 3.3, where the constant must be the same for any $f$, right?
* Under Equation 14, shouldn’t the last step be $\lim_{K\rightarrow\infty} \log K$ instead of $\log K$? If so, does Corollary 3.3 hold when constants are infinite?
* It seems the theory is inspired by Ghosh et al. [5], Equation 6 seems to be directly related to Wang & Isola [6], etc. Could you clearly state what parts of your theory are based on other work?
* In Section 3.4.1, it is claimed that InfoNCE-NN is noise tolerant. The argument seems to be that for each $f$, one can choose a specific threshold $t$ such that $\Delta \mathcal{R} (\mathcal{L}; f)$ becomes a constant. However, isn’t it so that Corollary 3.3 requires that the constant is chosen independent of f?


**General Clarifications.**

* If the RINCE loss is noise tolerant, then what motivates you to propose SymNCE? Does it in any way solve some issues of RINCE?
* Why not use positive (augmentations) and negative samples (other examples) with NT-Xent/InfoNCE as in say SimCLR [7] when using contrastive losses for label noise robustness? It seems using class information to choose positive and negative samples only lead to problems in the noisy labels case.

**References.**

[1] Wang Y, Ma X, Chen Z, Luo Y, Yi J, Bailey J. Symmetric cross entropy for robust learning with noisy labels

[2] Li J, Socher R, Hoi SC. Dividemix: Learning with noisy labels as semi-supervised learning.

[3] Liu S, Niles-Weed J, Razavian N, Fernandez-Granda C. Early-learning regularization prevents memorization of noisy labels.

[4] Yi L, Liu S, She Q, McLeod AI, Wang B. On learning contrastive representations for learning with noisy labels.

[5] Ghosh A, Kumar H, Sastry PS. Robust loss functions under label noise for deep neural networks.

[6] Wang T, Isola P. Understanding contrastive representation learning through alignment and uniformity on the hypersphere.

[7] Chen T, Kornblith S, Norouzi M, Hinton G. A simple framework for contrastive learning of visual representations.

---

### Official Review · Reviewer_PcdE · 2023-11-03

**Soundness:** 3 good
**Presentation:** 3 good
**Contribution:** 2 fair
**Rating:** 3
**Confidence:** 4

**Summary:**

The paper investigates the robustness of supervised contrastive loss for learning noisy labels. The paper aims to fill the gap between empirical utilization of supervised contrastive learning loss and its theorectical foundation. Specifically, the paper claims to be the first to derive a general robust condition for arbitrary contrastive losses. The framework can emcompass the nearest-neighbor sample selection and robust contrastive loss. Based on the theorecticaly analysis, the paper proposes a RevNCE loss which forms a Symmetric InfoNCE loss with the original InfoNCE loss. The paper evaluates the framework on CIFAR10&100 and Clothing1M.

**Strengths:**

1. The paper proposes a unified theorectical framework for robust supervised contrastive learning loss, which can apply to nearest neighbor sample selection and RINCE loss.

2. The proposed RevNCE is simple and easy to implement and does not add a large computational overhead.

**Weaknesses:**

1. The paper does not compare with latest contrastive learning based noisy label methods. [1, 2]. The proposed method's performance is not strong compared with latest contrastive learning methods.

2. The paper does not discuss the limitations of the proposed method. However, the assumptions of blanced class and symmetric noise are very strong in practice.

3. Specifically, I think most of current noisy label learning methods tackle the instance-dependent noise. Thus, the noise is not uniform among classes. The assumption 3.1 does not hold.

Meanwhile, using a unified threshold parameter t to determine positive examples is not reasonbale since the situation for each class may be different. And the clustering structure essentially may not have good quality.

In conclusion, I think symmetric label noise and class balance are not very pratical and reasonable assumption in the current noisy lable learning senario.

4. The experiment part lack results for instance-dependent noise.

5. There are some works related to contrastive learning on noisy label learning which are not well discussed and compared.

[1] Selective-supervised contrastive learning with noisy labels. CVPR 2022

[2] Twin Contrastive Learning with Noisy Labels. CVPR2023

[3] Beyond synthetic noise: Deep learning on controlled noisy labels. ICML 2020

[4] Learning with feature-dependent label noise: A progressive approach. ICLR 2021

[5] Investigating Why Contrastive Learning Benefits Robustness against Label Noise. ICML 2022

[6] Ngc: A unified framework for learning with open-world noisy data. ICCV 2021

**Questions:**

Please see the weakness part.

---

### Official Review · Reviewer_zzjJ · 2023-11-09

**Soundness:** 3 good
**Presentation:** 2 fair
**Contribution:** 2 fair
**Rating:** 3
**Confidence:** 4

**Summary:**

Contrastive Learning is a powerful tool to learn rich representations from training data via attractive and repulsive forces between semantically similar and dissimilar examples. Recently supervised contrastive learning (sCL) extend this paradigm to the supervised setting where by using the semantic annotations to pull together multiple positive pairs (anchor is attracted to multiple examples in the batch with same annotation). However, since sCL relies on semantic annotations (instead of only self augmentations) to form positive pairs, it is also sensitive to noisy labels. This paper, derives a general robust condition for arbitrary contrastive losses, which serves as a criterion to verify the theoretical robustness of a supervised contrastive loss against label noise. They propose a robust version of infoNCE loss they call Symmetric InfoNCE (SymNCE). They provide empirical evidence justifying their approach.

**Strengths:**

* Recent papers have undertaken the problem of studying contrastive losses under the robustness setup e.g. against noisy views (in the unsupervised setting) or against label noise (in the supervised setting) using the classical symmetric loss idea [1]. This paper unifies these approaches under a common theory framework.

* The problem and approach is well motivated from classic robustness theory.

**Weaknesses:**

* **Novelty**: Overall, I am not sure why this approach is novel. sCL is a supervised loss and naturally vulnerable to label noise. Many tricks are known (for non-contrastive setting) to robustify supervised losses against label noise. One such trick is to make the loss adhere to the symmetry condition. This trick is also well studied in PU Learning setting (a special label noise setting).

* Theorem 3.2 for example is a straightforward adaptation of Theorem 1 form [1]

* What happens when num of views M=1 i.e. we only have a single positive like SimCLR ~ I assume there is no way to recover if the augmentation is noisy view? This needs to be discussed in depth.

*  Or in other words The theoretical guarantee of the proposed loss holds when noise rate >= 1/2 ? I believe it should not ~~ as 1/2 is the theoretical limit of breakdown point of an estimator. However, I am surprised how at 80% noise level RINCE works so well ?? I feel it is because the way you create noisy dataset. If the flipped label is very far from the original class e.g instead of flipping dogs with cats => replace dogs with something orthogonal like say banana ... i dont think the empirical results would hold. Can you present results where you show what happens when you try different flips i.e. replace label i with an arbitrary label => not cat with dog only => but show for other flips too.

* You briefly mention works e.g. https://arxiv.org/pdf/2201.12498.pdf where they show why unsupervised CL is useful in being robust. Intuitively this makes sense --- since we are in label noise setting -- a natural baseline would be to not use sCL -> rather SimCLR should work better as it is agnostic to label noise.

* Also, it is necessary to compare the embeddings learnt via different loss => for example using SimCLR, sCL only learn the encoder and evaluate using supervised kNN or Linear Probing and investigate why SimCLR is not a good choice.

* There are other tricks used in noisy label setting -- for example see https://arxiv.org/pdf/2206.01206.pdf and  https://arxiv.org/abs/2306.04160 where a judicious combination of SimCLR and sCL works fine - may be you should consider discussing them in related work. Further, I think https://arxiv.org/abs/2306.04160 needs to be compared against too --- as this is super simple idea and should be strong baseline.

* Noise Assumption weaker ? - how about sample dependent noise ?

* The paper needs to be written better -- The current presentation is quite hard to read. The main contributions are unclear until Sec 4.

* Finally, I would like to see the code for creating the label noise (or give pointer to some existing work's github you used it from ) --- I feel the empirical success is highly dependent on which labels are flipped with which label and how .... Please consider sharing the code snippet about how you create the noisy dataset to ensure reproducibility.


1. [A. Ghosh, N. Manwani, and P. Sastry, “Making risk minimization tolerant to label noise,” Neurocomputing]

**Questions:**

Please refer to weakness

---

### Meta-Review · Area_Chair_1vGp · 2023-12-05

**Metareview:**

The paper discusses the robustness of the contrastive loss to noisy labels from a theoretical standpoint and proposes a modification via a Symmetric InfoNCE loss. There are several concerns about 1) novelty, 2) weakness/insufficiency of the experimental evidence and unfair comparisons, and 3) clarity of the theoretical results. Since the authors do not provide a response, I am proposing rejection.

**Justification For Why Not Higher Score:**

The concerns raised by the reviewers have not been addressed by the authors during the discussion period. Therefore, I cannot recommend a higher score at this state.

**Justification For Why Not Lower Score:**

N/A

---

### Decision · Program_Chairs · 2024-01-16

Reject